# Steer Where It Matters: Token-Level Visual-Sensitivity Steering for LVLMs Hallucination Mitigation

**Ruipeng Zhang** [1 2]  **Zhihao Li** [1 3 2]  **C. L. Philip Chen** [1 3 2]  **Tong Zhang** [1 3 2]

## Abstract

Large vision language models (LVLMs) have made rapid advancements and are deployed across various applications, yet hallucinations remain a major challenge. Activation steering is appealing due to its minimal training overhead and controllability at inference time. However, we found that during autoregressive decoding, visual conditioning affects token prediction sparsely and locally across decoding steps, and many existing methods that average image-versus-no-image differences over the entire sequence dilute these critical signals, yielding low signal-to-noise ratio steering directions. Additionally, many existing methods apply a fixed steering strength, which misallocates the intervention budget, over-perturbs non-critical tokens, and can cause instability. To address these limitations, we propose **Token-Level Visual-Sensitivity Steering** (TLVS) for hallucination mitigation. Our approach first extracts token-level steering vectors and refines them, and then applies fine-grained, visual-sensitivity–adaptive steering only where it matters. This lightweight, plug-and-play mechanism requires only minimal training for calibration and can be applied across diverse vision-language models. It modulates the steering strength at each decoding step, selectively suppressing hallucination-prone spans while preserving evidence-grounded content. We evaluate TLVS on several benchmarks, including POPE, AMBER, CHAIR (COCO), MMHal and HallusionBench, demonstrating consistent improvements over previous steering methods.

[1]Guangdong Provincial Key Laboratory of Computational AI Models and Cognitive Intelligence, School of Computer Science & Engineering, South China University of Technology, Guangzhou, China [2]Engineering Research Center of the Ministry of Education on Health Intelligent Perception and Paralleled Digital-Human, Guangzhou, China [3]Pazhou Lab, Guangzhou, China. Correspondence to: Tong Zhang <tony@scut.edu.cn>.

*Proceedings of the 43rd International Conference on Machine Learning*, Seoul, South Korea. PMLR 306, 2026. Copyright 2026 by the author(s).

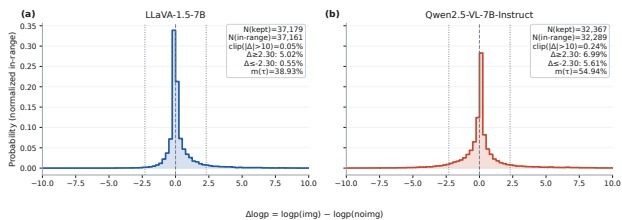

*Figure 1.* Distribution of $\Delta \log p$ in Eq. (3) under teacher forcing for LLaVA-1.5-7B and Qwen2.5-VL-7B-Instruct. Most tokens cluster near $\Delta \log p \approx 0$, while a small heavy-tailed fraction accounts for a disproportionate share of the total visual gain.

## 1. Introduction

Large vision language models (LVLMs) have shown impressive performance in vision-language understanding and generation (Li et al., 2022; Liu et al., 2023; Lu et al., 2024; Liu et al., 2024a; Bai et al., 2025), enabling applications such as assistants, embodied agents, and content creation in healthcare (Huang et al., 2023; Zitkovich et al., 2023; Qian et al., 2025). Nevertheless, hallucinations remain a major bottleneck for LVLMs, often attributed to an over-reliance on language priors, potentially stemming from architectural and training biases (Leng et al., 2024; Zhu et al., 2025; Fang et al., 2025). Existing methods broadly include training-time alignment and inference-time control. Training-time approaches (Yu et al., 2024; Xie et al., 2024; Wang et al., 2024; Zhang et al., 2026) can effectively mitigate hallucinations by reducing over-reliance on language priors. They are often computationally expensive and time-consuming to train. Meanwhile, a growing body of work explores inference-time interventions, spanning contrastive decoding and attention modulation (Leng et al., 2024; Liu et al., 2024b; Jung et al., 2025; Tang et al., 2025; Huo et al., 2024; Yang et al.). Among them, activation steering (Li et al., 2025; Liu et al., 2025) has recently gained attention as it requires little to no training and can be readily plugged into existing LVLMs by injecting a learned direction into the model's hidden states. The dominant paradigm constructs two conditions (image-conditioned vs. image-ablated), extracts a "more-visual" direction by aggregating their activation differences, and applies the direction globally with a fixed steering strength during generation. However, this **token-agnostic** and **step-invariant** formulation implicitly

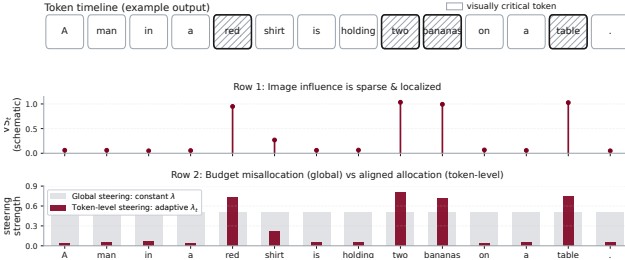

*Figure 2.* Visual evidence matters at only a few steps (spikes in $VS_t$), while global steering applies a constant $\lambda$ everywhere, over-perturbing image-insensitive tokens. Token-level steering adapts $\lambda_t$ to $VS_t$, focusing intervention on visually critical steps.

assumes that visual evidence contributes uniformly across decoding steps and requires the same steering strength at every step. This raises a natural question: when and where does visual grounding truly matter during generation, and how should steering budget be allocated accordingly? Our analysis provides two observations that motivate a more fine-grained alternative.

Firstly, as Figure 1 shows, the token-level visual gain $\Delta \log p$ has a sharp peak near zero with heavy tails. This indicates that the output logits change little for most decoding steps, while a small fraction of tail tokens accounts for a substantial share of the overall visual influence and undergoes large changes. This suggests that global averaging can dilute the critical visual signals. Secondly, as Figure 2 illustrates, global steering applies a uniform intervention across all steps, even though visual reliance is highly sparse across time, as captured by spikes in the token-level visual sensitivity signal $VS_t$, defined in Eq. (17). As a result, it inevitably perturbs many non-critical tokens and misallocates the intervention budget. We formalize these two problems in Sec. 3.

Motivated by this diagnosis, we propose Token-Level Visual-Sensitivity Steering (TLVS) that steers only where it matters. Concretely, (i) we extract per-layer steering directions by contrasting token representations at visually sensitive vs. insensitive steps; (ii) we optionally refine these directions using human-corrected responses with a lightweight objective that denoises them while regularizing distribution drift; and (iii) we perform token-level adaptive steering that modulates the steering strength step-by-step using an online visual-sensitivity signal, with smoothing and confidence-based caps for stability. We evaluate TLVS on two widely used LVLMs (LLaVA-1.5-7B and Qwen2.5-VL-7B) across standard hallucination benchmarks (POPE, AMBER, MMHal-Bench, HallusionBench, and CHAIR (COCO)). TLVS consistently reduces hallucinations while improving stability and largely preserving overall generation quality. Our contributions are as follows.

- We provide an empirical diagnosis: image condition-

ing influences token prediction sparsely with a heavy-tailed distribution, while step-invariant global steering unnecessarily perturbs many non-critical tokens.

- We propose **Token-Level Visual-Sensitivity Steering (TLVS)**, which first extracts and refines token-level steering vectors, and then applies fine-grained, visual-sensitivity-adaptive steering only where it matters.

- We demonstrate consistent reductions in hallucinations with minimal collateral disturbance on LLaVA-1.5-7B and Qwen2.5-VL-7B-Instruct across POPE, AMBER, MMHal-Bench, HallusionBench, and CHAIR (COCO).

## 2. Related Work

Hallucination in large vision language models (LVLMs) refers to generating content that is unsupported by the visual input, often driven by strong language priors and imperfect cross-modal grounding that can produce fluent but visually unfaithful outputs (Zhou et al., 2023; Li et al., 2023; Guan et al., 2024; Cao et al., 2024).

**Hallucination mitigation in LVLMs.** Existing methods broadly fall into training-time alignment and inference-time control. Training-time approaches improve grounding through instruction tuning (You et al., 2023; Yuan et al., 2024; Garg et al., 2024) and preference optimization with curated or corrective supervision. Representative DPO-based variants collect image-conditioned preference pairs and directly optimize the model to favor visually grounded outputs without training an explicit reward model (Rafailov et al., 2023; Yu et al., 2024; Wang et al., 2024; Xing et al., 2025). Inference-time methods are attractive for their plug-and-play deployment. A major line reshapes decoding with contrastive/calibrated schemes that subtract or reweight logits using perturbed or auxiliary branches, suppressing language-prior continuations and promoting image-grounded tokens (Huo et al., 2024; Leng et al., 2024; Chen et al., 2025; Zhang et al., 2025a). Others calibrate attention or confidence to downweight (Zhang et al., 2025b; Wu et al., 2024) overconfident but unsupported tokens, often with stepwise heuristics that adapt intervention strength during generation (Jung et al., 2025). Complementary approaches perform post-hoc verification and revision (Zhou et al., 2023; Yin et al., 2024; Lee et al., 2024a;b), using the model itself or an auxiliary verifier to identify and correct ungrounded spans at the cost of extra computation.

**Activation steering.** Activation steering manipulates intermediate representations at inference time to control model behavior. In language-only settings (Turner et al., 2024; Arditi et al., 2024; Rimsky et al., 2024), steering directions are typically derived from contrastive prompts and injected into hidden states to induce global behaviors (e.g., senti-

ment, safety, honesty), and some of them further studie how such directions can be identified and composed (Stolfo et al., 2024; Vu & Nguyen, 2025). Recent multimodal work has begun to explore steering-style interventions for hallucination mitigation in LVLMs (Li et al., 2025; Liu et al., 2025; Wu et al., 2025; Su et al., 2025). Most methods aggregate image–no-image activation differences over the whole sequence, diluting sparse visual signals and producing low-SNR directions. Their fixed steering strength further over-perturbs non-critical tokens, causing instability. In contrast, we formulate LVLMs' hallucination mitigation as a token-level control problem: TLVS first extracts and refines token-level steering vectors, and then applies fine-grained, visual-sensitivity–adaptive steering only where it matters.

## 3. Problem Formulation and Diagnosis

Activation steering is a plug-and-play, inference-time control method for hallucination mitigation. A common paradigm contrasts image-conditioned with image-ablated inputs (Li et al., 2025; Liu et al., 2025), pools their activation differences to obtain a "more-visual" direction, and applies it uniformly during decoding.

Let $h_{\ell,t}^I(x)$ and $h_{\ell,t}^\varnothing(x)$ denote layer-$\ell$ hidden states at step $t$ under the image-conditioned and image-ablated inputs. A standard choice estimates a layer-wise direction by token-averaging the condition difference:

$$v^{(\ell)} = \mathbb{E}_{x \sim \mathcal{D}}\left[\frac{1}{T}\sum_{t=1}^{T}\left(h_{\ell,t}^I(x) - h_{\ell,t}^\varnothing(x)\right)\right], \qquad \ell \in \mathcal{S}, \tag{1}$$

At inference time, a common baseline applies step-invariant injection:

$$h'_{\ell,t} = h_{\ell,t} + \lambda\, v^{(\ell)}, \qquad \ell \in \mathcal{S}, \tag{2}$$

where $\lambda$ is the steering strength.

However, this raises two concerns: (i) the effect of visual conditioning on token prediction may be sparse and localized, making coarse extraction insufficiently targeted; and (ii) globally applying a fixed intervention may perturb many tokens that do not require additional visual grounding, leading to systematic side effects and decoding instability. These considerations motivate a closer look at token-level selection and adaptive modulation of the intervention budget across decoding steps.

### 3.1. Observation 1: Image conditioning affects token prediction sparsely

To quantify how much the image condition changes token prediction, we compute token-level differences using teacher forcing so that the two conditions are compared under an identical target sequence (see Appendix A).

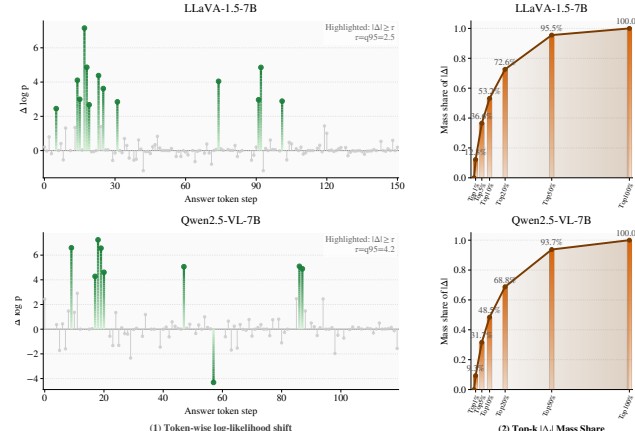

*Figure 3.* Left: per-token $\Delta_t$ on one example, where large-magnitude tokens occur sparsely. Right: mass concentration of $|\Delta_t|$, showing that a small top fraction dominates the total.

Concretely, for each example we first fix an answer token sequence $y_{1:T}$ by taking the model-generated response under the image-conditioned setting. We then compute the step-wise token log-probabilities under the two conditions with an aligned prefix, and take their difference:

$$\Delta_t = \log p(y_t \mid x, I, y_{<t}) - \log p(y_t \mid x, \varnothing, y_{<t}), \tag{3}$$

where $I$ denotes the image input and $\varnothing$ the image-ablated condition. Importantly, Eq. (3) is computed with the same prefix $y_{<t}$, ensuring that $\Delta_t$ isolates the effect of visual conditioning.

Figure 3 illustrates representative lollipop plots of $\Delta_t$ along decoding steps. We observe a clear sequence-level sparsity pattern: most tokens have $\Delta_t \approx 0$, while only a small subset exhibits large deviations, typically corresponding to visually grounded. This indicates that visual information does not uniformly influence the entire generation; rather, it intervenes at a small subset of steps.

To quantify this pattern at the dataset level, we aggregate all answer tokens across the dataset, sort them by $|\Delta_t|$ in descending order, and measure how much cumulative $|\Delta_t|$-mass is captured by the top fraction of tokens:

$$S(p) = \frac{\sum_{t \in \text{Top}(p)} |\Delta_t|}{\sum_t |\Delta_t|}, \tag{4}$$

where $\text{Top}(p)$ denotes the set of the top $p$ fraction of tokens ranked by $|\Delta_t|$. As shown in Figure 3, $|\Delta_t|$ is highly concentrated: a small top fraction of tokens already explains a disproportionately large share of the total $|\Delta_t|$-mass, and the cumulative curve rises steeply from 0% to 100%. In particular, the bars at $p \in \{1, 5, 10, 20, 50, 100\}\%$ show that the $|\Delta_t|$-mass accumulates rapidly within the top-ranked tokens, indicating a strongly top-heavy pattern. Figure 4 further shows that, across the dataset, most tokens cluster near $\Delta_t \approx 0$, with only sparse large-magnitude deviations spread

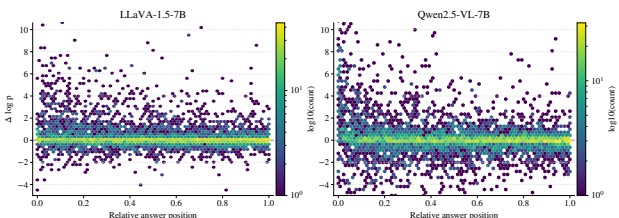

*Figure 4.* Hexbin density plots show $\Delta_t$ for individual tokens as a function of their relative answer position; most tokens cluster near zero while a small fraction exhibits large-magnitude deviations. Color indicates log-scaled counts.

over relative answer positions. Together, these results indicate that image conditioning is sparse across decoding steps and dominated by a small set of high-magnitude tokens.

This observation has an important implication for steering signal extraction. When the image-induced effect concentrates on a small subset of steps, uniformly averaging activations over all tokens and then taking a condition difference can wash out these critical contributions, yielding a low-SNR direction. We formalize this dilution effect below.

Let $\delta_{\ell,t}(x) = h^I_{\ell,t}(x) - h^{\varnothing}_{\ell,t}(x)$ be the layer-$\ell$ activation difference at step $t$, and let $\mathcal{C}_\tau = \{t : |\Delta_t| \geq \tau\}$ denote visually critical steps with sparsity rate $q = |\mathcal{C}_\tau|/T$. Under a simple sparse-effect model in which only $t \in \mathcal{C}_\tau$ carries a consistent mean shift $v^{(\ell)}_\star$ (and non-critical steps are zero-mean), the step-averaged estimator used by Eq. (1) satisfies

$$\mathbb{E}\Big[\frac{1}{T}\sum_{t=1}^{T} \delta_{\ell,t}(x)\Big] = q\, v^{(\ell)}_\star. \qquad (5)$$

Moreover, if one rescales this average to be unbiased, $\tilde{v}^{(\ell)}_{\text{avg}} = \frac{1}{qT}\sum_{t=1}^{T} \delta_{\ell,t}(x)$, its estimation error is worse by a factor on the order of $1/q$ compared to selecting/averaging only over critical steps, yielding a low-SNR direction (see Appendix B). Finally, without token alignment, free-running generations can introduce additional path-divergence noise into $\delta_{\ell,t}(x)$, further reducing the effective SNR.

### 3.2. Observation 2: Step-invariant global steering over-perturbs non-critical tokens

Many activation-steering baselines are step-invariant, applying the same direction with a constant strength at every decoding step (Eq. (2)). To examine the limitation of such step-invariant application, we run a teacher-forced, token-aligned diagnosis on a fixed response $y_{1:T}$ and compare how different intervention schedules affect the likelihood of the same reference tokens. Implementation details are deferred to Appendix C. Specifically, for each token step $t$, we measure the per-token change in negative log-likelihood (NLL) relative to vanilla decoding:

$$\Delta\text{NLL}_t(\text{method} - \text{vanilla}) = \text{NLL}^{\text{method}}_t - \text{NLL}^{\text{vanilla}}_t, \quad (6)$$

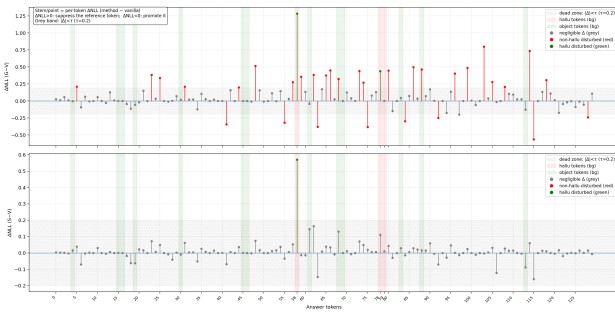

*Figure 5.* Teacher-forced, token-aligned $\Delta\text{NLL}_t$ for global vs. oracle localized steering (Eq. 6). Grey band: $|\Delta_t| < \tau$; highlights: AMBER hallucinated (red) and object-grounded (green) spans. Global steering perturbs many background tokens, while localization focuses on critical tokens.

where $\Delta\text{NLL}_t > 0$ suppresses the reference token $y_t$ (lower probability) and $\Delta\text{NLL}_t < 0$ promotes it. Figure 5 contrasts two methods using the same steering direction: (i) global step-invariant steering applied at every token step, and (ii) an oracle token-aware schedule that uses AMBER annotations to assign a larger strength to visually critical steps (e.g., hallucination and object-related tokens) and a smaller strength otherwise. The top panel shows that global steering induces substantial $|\Delta\text{NLL}_t|$ at many non-critical steps (i.e., non-hallucinated, correct/background tokens), and can also perturb evidence-grounded spans. In contrast, the oracle schedule concentrates $\Delta\text{NLL}_t$ on visually critical tokens and suppresses background disturbances, suggesting that step-invariant steering mainly suffers from misallocated intervention budget. This motivates token-level adaptive modulation of steering strength per step to focus on visually critical (hallucination-prone) tokens.

## 4. Method

We consider an LVLM under two conditions: an image-conditioned input $(x, I)$ and an image-ablated input $(x, \varnothing)$, where the image is removed. The model defines the next-token distribution $p_\theta(\cdot \mid x, I, y_{<t})$ (and $p_\theta(\cdot \mid x, \varnothing, y_{<t})$ analogously).

We intervene on a set of transformer layers $\mathcal{S}$ by adding a steering vector to the hidden states:

$$h^{(\ell)}_t \leftarrow h^{(\ell)}_t + \lambda_t W^{(\ell)}, \qquad \forall \ell \in \mathcal{S}, \qquad (7)$$

where $W^{(\ell)} \in \mathbb{R}^d$ is the steering direction for layer $\ell$ and $\lambda_t \geq 0$ is a (possibly time-varying) steering strength.

With this setup, our method has three components: (1) token-level extraction of per-layer steering vectors $W^{(\ell)}_{\text{init}}$, (2) optional supervised refinement of $W^{(\ell)}$, and (3) KL-controlled adaptive steering at inference time, which adapts the steering strength $\lambda_t$ token by token. Figure 6 provides an overview.

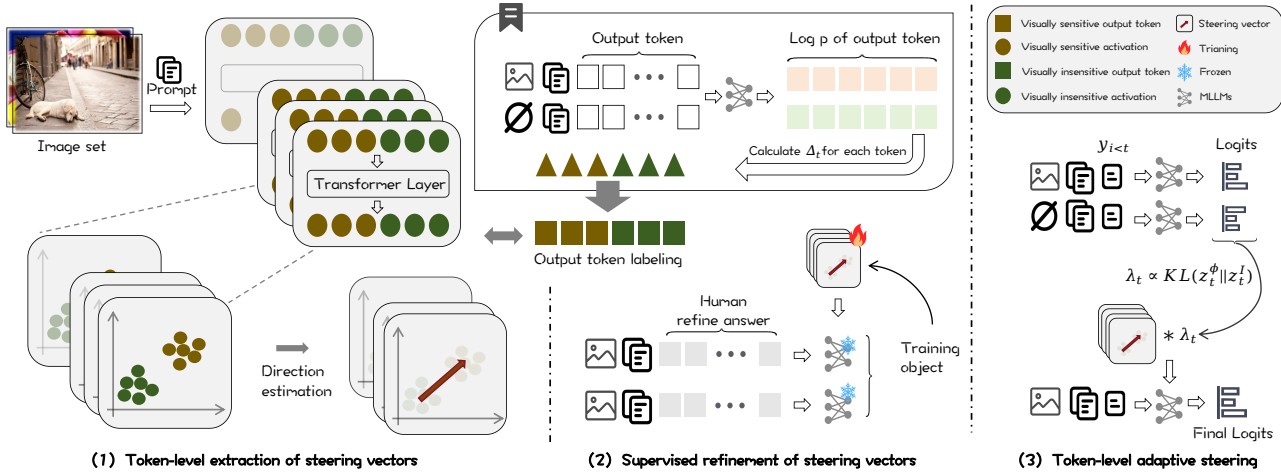

*Figure 6.* Overview of our token-level steering framework. Step 1 extracts per-layer steering directions from token-wise image sensitivity ($\Delta_t$); Step 2 optionally refines $W^{(\ell)}$ with supervised correction data; Step 3 adaptively updates $\lambda_t$ at each step using the KL divergence between image-conditioned and image-ablated logits.

## 4.1. Step 1: Token-level extraction of steering vectors

**Representation logging.** We first construct a calibration set. For each sample $(I, x)$, we obtain a reference answer $\hat{y} = (\hat{y}_1, \ldots, \hat{y}_T)$ by running the model conditioned on the image. We then record token-level hidden representations $\{h_t^{(\ell)}\}_{\ell=1..L,\ t=1..T}$ corresponding to $\hat{y}$. This yields token-indexed representations that we will partition into visually sensitive and insensitive subsets.

**Token labeling.** We estimate how much each token depends on the image by comparing teacher-forced log-probabilities with and without the image input. Specifically, for each step $t$ we compute:

$$\Delta_t = \log p_\theta(\hat{y}_t \mid x, I, \hat{y}_{<t}) - \log p_\theta(\hat{y}_t \mid x, \varnothing, \hat{y}_{<t}). \quad (8)$$

We interpret $|\Delta_t|$ as the magnitude of visual dependence at step $t$: larger $|\Delta_t|$ indicates that the token likelihood is more strongly affected by the image condition. The sign of $\Delta_t$ indicates whether the image increases ($\Delta_t > 0$) or decreases ($\Delta_t < 0$) the likelihood of the reference token.

Let $\mathcal{T}$ be the set of valid answer-token indices. We form pseudo-labels by selecting extreme subsets under $|\Delta_t|$:

$$\begin{aligned} \mathcal{T}^+ &= \text{TOP}_\rho\big(\{|\Delta_t|\}_{t\in\mathcal{T}}\big), \\ \mathcal{T}^- &= \text{BOTTOM}_\rho\big(\{|\Delta_t|\}_{t\in\mathcal{T}}\big). \end{aligned} \quad (9)$$

where $\rho \in (0, 1)$ is a fraction (percentage) of tokens. Here $\mathcal{T}^+$ denotes visually sensitive tokens (large $|\Delta_t|$) and $\mathcal{T}^-$ denotes visually insensitive tokens (small $|\Delta_t|$, i.e., closest to 0). We use these pseudo-labels to estimate per-layer directions that preferentially increase image reliance at the corresponding decoding steps.

**Direction estimation.** For each layer $\ell$, we L2-normalize token representations:

$$\tilde{h}_t^{(\ell)} = \frac{h_t^{(\ell)}}{\|h_t^{(\ell)}\|_2 + \epsilon}, \quad (10)$$

and compute a per-sample difference-of-means vector:

$$d^{(\ell)} = \frac{1}{|\mathcal{T}^+|} \sum_{t\in\mathcal{T}^+} \tilde{h}_t^{(\ell)} - \frac{1}{|\mathcal{T}^-|} \sum_{t\in\mathcal{T}^-} \tilde{h}_t^{(\ell)}. \quad (11)$$

We aggregate $\{d^{(\ell)}\}$ across calibration samples and take the first principal component to obtain the initial steering vector:

$$W_{\text{init}}^{(\ell)} = \text{PCA}_1\big(\{d_i^{(\ell)}\}_{i=1}^N\big), \qquad W_{\text{init}}^{(\ell)} \leftarrow \frac{W_{\text{init}}^{(\ell)}}{\|W_{\text{init}}^{(\ell)}\|_2}. \quad (12)$$

Intuitively, moving in the direction $W_{\text{init}}^{(\ell)}$ shifts token representations toward those that are more affected by image conditioning.

## 4.2. Step 2: Supervised refinement of steering vectors

After estimating initial per-layer steering directions $W_{\text{init}}^{(\ell)}$ from the pseudo-labeled token sets $\mathcal{T}^+$ and $\mathcal{T}^-$, we refine these directions using supervised image–question–human-corrected answer triples from the RLHF-V dataset (Yu et al., 2024), where human annotators correct hallucinated segments in a model-produced response.

We optimize only the steering vectors $W^{(\ell)}$ for $\ell \in \mathcal{S}$. During training, we apply a fixed steering strength $\lambda$ in Eq. (7).

**Paired teacher forcing (reference vs. steered).** For each training sample $(I, x, y)$, we run two teacher-forcing passes under the *same* image input: (i) a reference pass with steering disabled, yielding $p_\theta(\cdot \mid x, I, y_{<t})$; and (ii) a steered pass with steering enabled, yielding $p_{\theta,W}(\cdot \mid x, I, y_{<t})$, where gradients flow only to $W^{(\ell)}$. This design isolates the effect of steering while holding the image input fixed.

We optimize the following objective on answer tokens $\mathcal{A}$:

$$\mathcal{L} = \mathcal{L}_{\text{NLL}} + \beta \, \mathcal{L}_{\text{KL}} + \gamma \, \mathcal{L}_{\text{prox}}. \qquad (13)$$

The negative log-likelihood over answer tokens is:

$$\mathcal{L}_{\text{NLL}} = -\frac{1}{|\mathcal{A}|} \sum_{t \in \mathcal{A}} \log p_{\theta,W}(y_t \mid x, I, y_{<t}). \qquad (14)$$

To limit distribution shift, we regularize with a trust-region KL, optionally approximated on the top-$K$ probability mass with renormalization for efficiency.

$$\mathcal{L}_{\text{KL}} = \frac{1}{|\mathcal{A}|} \sum_{t \in \mathcal{A}} \text{KL}(p_{\theta,W}(\cdot \mid x, I, y_{<t}) \,\|\, p_\theta(\cdot \mid x, I, y_{<t})). \qquad (15)$$

Finally, we apply a proximal penalty anchoring vectors to their initialization:

$$\mathcal{L}_{\text{prox}} = \frac{1}{|\mathcal{S}|} \sum_{\ell \in \mathcal{S}} \left\| W^{(\ell)} - W_{\text{init}}^{(\ell)} \right\|_2^2. \qquad (16)$$

### 4.3. Step 3: Visual-sensitivity–adaptive steering

We perform stepwise decoding and adapt the steering strength $\lambda_t$ using a KL-based visual sensitivity signal computed from a dual-route forward process. The key idea is to allocate stronger steering to steps that exhibit high dependence on the image.

At each decoding step $t$, we compute logits from two routes: (i) an image-conditioned route producing logits $z_t^I = f_\theta(x, I, y_{<t})$; and (ii) a no-image route producing logits $z_t^\varnothing = f_\theta(x, \varnothing, y_{<t})$. We define the visual sensitivity score as:

$$\text{VS}_t = \text{KL}\left( \text{softmax}\left( \frac{z_t^I}{T_{KL}} \right) \,\middle\|\, \text{softmax}\left( \frac{z_t^\varnothing}{T_{KL}} \right) \right), \qquad (17)$$

where $T_{KL}$ is a temperature used for the KL computation. For efficiency, we optionally approximate the KL on a truncated top-$K$ vocabulary (selected by the image-conditioned route and renormalized). We standardize the score using calibration statistics $(\mu, \sigma)$ computed on a held-out calibration set:

$$\bar{\text{VS}}_t = \frac{\text{VS}_t - \mu}{\sigma + \epsilon}. \qquad (18)$$

We map $\bar{\text{VS}}_t$ to a soft gate via a sigmoid:

$$g_t = \text{sigmoid}\left( \frac{\bar{\text{VS}}_t - b}{s} \right), \qquad (19)$$

and convert it to a steering strength within $[\lambda_{\min}, \lambda_{\max}]$:

$$\tilde{\lambda}_t = \lambda_{\min} + (\lambda_{\max} - \lambda_{\min}) \, g_t. \qquad (20)$$

To reduce stepwise jitter, we optionally apply exponential smoothing:

$$\hat{\lambda}_t = \alpha \, \hat{\lambda}_{t-1} + (1 - \alpha) \, \tilde{\lambda}_t. \qquad (21)$$

We set $\hat{\lambda}_0 = \lambda_{\min}$. We further constrain the steering strength using a per-step cap $\lambda_t^{\text{cap}}$, derived from a confidence proxy of the image-conditioned logits $z_t^I$ (see Appendix E), and take:

$$\lambda_t = \min(\hat{\lambda}_t, \lambda_t^{\text{cap}}). \qquad (22)$$

## 5. Experiments

We evaluate TLVS in a staged manner: (i) we first report main results on standard benchmarks to assess overall effectiveness (Section 5.2); (ii) we conduct ablation studies to isolate the contributions of TLVS's refinement and adaptive steering mechanisms (Section 5.3); (iii) we provide mechanistic and qualitative analyses to verify that TLVS allocates intervention budget to hallucinated spans at the token level (Section 5.4); and (iv) we measure inference-time overhead to quantify efficiency and practicality (Section 5.5).

### 5.1. Experimental Setup

**Models and Data.** We conduct our main experiments on LLaVA-1.5-7B (Liu et al., 2024a), a widely used open-source LVLM backbone. To verify the generality of TLVS on a newer and stronger architecture, we additionally report results on Qwen2.5-VL-7B (Bai et al., 2025). Both the extraction of activation steering vectors and the optional supervised refinement use examples from the RLHF-V correction dataset (Yu et al., 2024).

**Benchmarks.** We evaluate TLVS on five widely adopted hallucination benchmarks covering both probing-style and open-ended generation settings: POPE (Li et al., 2023), AMBER (Wang et al., 2023), HallusionBench (Guan et al., 2024), MMHal-Bench (Sun et al., 2024), and COCO Captioning with CHAIR metrics (Rohrbach et al., 2018). These benchmarks collectively capture object existence and attribute errors, open-ended description faithfulness, and multimodal reasoning hallucinations.

**Baselines.** For a fair comparison, we evaluate TLVS against the base model and a diverse set of representative hallucination mitigation baselines, covering both inference-time control (predominantly training-free) and training-time alignment. Among training-free methods that do not explicitly inject activation-steering directions, we include

*Table 1.* Main results. TLVS consistently improves hallucination-related metrics across POPE, HallusionBench, CHAIR (COCO), and AMBER on both LLaVA-1.5-7B and Qwen2.5-VL-7B. ↑/↓ indicate that higher/lower is better; **bold**/underline mark the best/second-best within each backbone; gray highlights TLVS, with ↑ / ↓ showing absolute gains over the vanilla baseline.

| Model | POPE | | | | HallusionBench | | | CHAIR (COCO) | | | AMBER | | | | | | | |
|---|---|---|---|---|---|---|---|---|---|---|---|---|---|---|---|---|---|---|
| | Adv.↑ | Pop.↑ | Rand.↑ | Avg.↑ | qAcc↑ | fAcc↑ | aAcc↑ | CHAIRs↓ | CHAIRi↓ | Recall↑ | CHAIR↓ | Cover↑ | Hal↓ | Cog↓ | F1-E↑ | F1-A↑ | F1-R↑ | F1↑ |
| LLaVA-1.5-7B | 81.80 | 84.36 | 89.12 | 85.09 | 13.19 | 16.18 | 45.70 | 46.0 | 12.8 | 78.9 | 12.0 | 50.3 | 36.4 | 4.6 | 83.2 | 65.6 | 62.4 | 74.7 |
| + VCD | 81.33 | 85.06 | 87.16 | 84.52 | 12.75 | 19.08 | 46.41 | 47.8 | 14.1 | **82.7** | 10.1 | 51.2 | 43.6 | 4.3 | 84.1 | 63.6 | 66.4 | 74.8 |
| + SPARC | 81.03 | 86.05 | 89.00 | 85.36 | 13.19 | 17.34 | 35.25 | 43.4 | 16.8 | 61.1 | 14.5 | 44.6 | 23.1 | 2.4 | 88.5 | 66.7 | 65.5 | 77.9 |
| + ASD | — | — | — | 87.87 | — | — | — | 40.0 | 11.3 | 82.0 | — | — | — | — | — | — | — | — |
| + SHARP | 79.00 | 83.20 | 85.30 | 82.50 | — | — | — | 34.8 | 10.6 | 59.7 | 8.5 | **52.1** | 39.2 | 4.8 | — | — | — | — |
| + VTI | 80.43 | 85.53 | 88.50 | 84.82 | 13.19 | 15.90 | 44.11 | 35.8 | 11.1 | 76.8 | 4.7 | 47.2 | 27.0 | 1.8 | 85.8 | 67.5 | 68.3 | 77.1 |
| + VISTA | 80.98 | 88.87 | 89.25 | 86.37 | 14.29 | 19.36 | 45.82 | 44.6 | 13.4 | 77.0 | 6.0 | 39.9 | 27.1 | 2.4 | 87.6 | 67.3 | 55.9 | 77.6 |
| + DPO (RLHF-V) | 84.03 | 85.94 | 89.12 | 86.36 | **16.70** | **20.81** | 51.31 | 43.2 | 11.9 | 76.4 | 5.7 | 49.7 | 27.3 | 2.6 | 90.7 | 72.6 | 64.6 | 80.9 |
| + TLVS (ours) | **87.85** ↑6.05 | **89.42** ↑5.06 | **92.08** ↑2.96 | **89.78** ↑4.69 | 15.85 ↑2.66 | 19.95 ↑3.77 | 46.50 ↑0.80 | **31.2** ↓14.8 | **9.7** ↓3.1 | 82.2 ↑3.3 | **3.8** ↓8.2 | 51.8 ↑1.5 | **19.7** ↓16.7 | **1.5** ↓3.1 | **91.5** ↑8.3 | **73.0** ↑7.4 | **69.7** ↑7.3 | **82.2** ↑7.5 |
| Qwen2.5-VL-7B | 84.92 | 86.78 | 89.01 | 86.90 | 23.74 | 31.21 | 55.54 | 24.0 | 7.0 | **86.7** | 5.4 | 51.6 | 29.0 | 1.9 | 97.3 | 83.8 | 72.6 | 89.4 |
| + VCD | 83.92 | 85.80 | 88.41 | 86.04 | 31.43 | 35.26 | 61.82 | 29.8 | 9.7 | 58.0 | 6.2 | 50.3 | 36.5 | 2.1 | 91.6 | 78.6 | 68.0 | 83.8 |
| + DPO (RLHF-V) | 84.99 | 86.81 | 89.72 | 87.17 | **34.95** | **39.60** | 62.62 | 13.4 | 3.5 | 62.7 | 3.1 | 50.7 | 21.9 | 1.1 | 98.5 | 82.5 | 65.9 | 88.3 |
| + TLVS (ours) | **85.10** ↑0.18 | **87.01** ↑0.23 | **89.88** ↑0.87 | **87.33** ↑0.43 | 33.08 ↑9.34 | 37.19 ↑5.98 | **63.23** ↑7.69 | **12.0** ↓12.0 | **5.2** ↓1.8 | 83.9 ↓2.8 | **3.8** ↓1.6 | 51.2 ↓0.4 | **16.7** ↓12.3 | **1.0** ↓0.9 | 97.3 | 82.7 ↓1.1 | 73.0 ↑0.4 | 89.8 ↑0.4 |

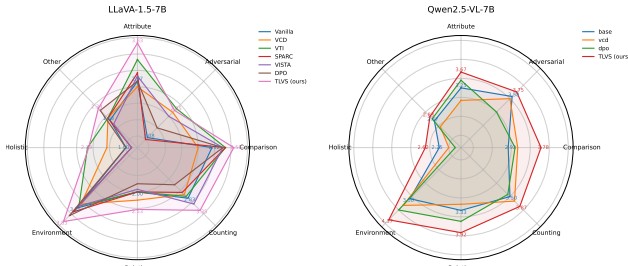

*Figure 7.* MMHal-Bench performance across question categories (attributes, adversarial objects, comparisons, counting, spatial, environmental, holistic, and others).

VCD (Leng et al., 2024) and SPARC (Jung et al., 2025). For activation-space steering approaches, we consider ASD (Su et al., 2025) and SHARP (Wu et al., 2025), as well as recent inference-time intervention methods VTI (Liu et al., 2025) and VISTA (Li et al., 2025). Finally, as a representative training-based baseline, we report results from RLHF-V-style behavior alignment (Yu et al., 2024) when applicable.

**Implementation details.** Unless otherwise noted, we use a unified prompting template and decoding configuration across methods for fair comparison. Full generation settings and implementation details are provided in Appendix Section D.

### 5.2. Main Results on Standard Benchmarks

As shown in Table 1, TLVS yields overall gains on both LLaVA-1.5-7B and Qwen2.5-VL-7B across POPE, HallusionBench, CHAIR (COCO), and AMBER. These benchmarks probe complementary hallucination settings, spanning object existence, open-ended captioning, diverse visual reasoning, and the factuality–coverage trade-off. Overall, TLVS achieves the best or second-best results on most metrics, demonstrating strong effectiveness and cross-backbone generalization.

TLVS consistently improves visual grounding across both

object-centric QA and open-ended captioning. On POPE, it increases average accuracy from 85.09 to 89.78 on LLaVA-1.5-7B and from 86.90 to 87.33 on Qwen2.5-VL-7B, with gains holding across all splits. On COCO captioning, TLVS further reduces hallucinated objects for both backbones: for LLaVA-1.5-7B, CHAIRs decreases from 46.0 to 31.2 and CHAIRi from 12.8 to 9.7, while Recall rises from 78.9 to 82.2; for Qwen2.5-VL-7B, hallucination-related metrics also improve, demonstrating consistent gains across model families.

Importantly, TLVS reduces hallucinations without substantially sacrificing coverage. For LLaVA-1.5-7B on AMBER, Hal drops from 36.4 to 19.7 and Cog from 4.6 to 1.5, while Cover increases from 50.3 to 51.8. On Qwen2.5-VL-7B, Hal decreases from 29.0 to 16.7 and Cog from 1.9 to 1.0, with Cover stable.

Finally, we further conduct an analysis over MMHal categories (Fig. 7) to comprehensively characterize TLVS's behavior across diverse question types. The radar plots indicate that TLVS improves performance consistently across diverse question types, rather than relying on gains from a narrow subset. This pattern is consistent with TLVS applying targeted, visually grounded interventions under multi-faceted stress tests.

### 5.3. Ablation Study

To isolate the contribution of each component, we compare three variants. **TLVS-Init** uses the initial vectors $W_{\text{init}}$ with a fixed steering strength $\lambda$, thereby testing whether the raw direction is already informative. **TLVS-Refine** replaces $W_{\text{init}}$ with the supervisedly refined vectors $W$, while keeping $\lambda$ fixed, isolating the effect of improving direction quality. **TLVS** is the full method, combining $W$ with token-level adaptive steering.

Table 2 isolates the roles of vector refinement and token-level adaptive steering. On LLaVA-1.5-7B, vector refinement (TLVS-Refine) improves POPE Avg. from 86.83 to

*Table 2.* Ablation on refinement and token-level adaptive steering. ↑ / ↓ indicate higher/lower is better; green/red arrows show absolute gains over the vanilla baseline within each backbone.

| Method | Components | | POPE | AMBER | | | |
|---|---|---|---|---|---|---|---|
| | Refine | Adapt. | Avg↑ | CHAIR↓ | Cover↑ | Hal↓ | F1↑ |
| **LLaVA-1.5-7B** | | | | | | | |
| Vanilla | – | – | 85.09 | 12.0 | 50.3 | 36.4 | 74.7 |
| TLVS-Init | × | × | 86.83 ↑1.74 | 7.4 ↓4.6 | 45.4 ↓4.9 | 35.9 ↓0.5 | 75.0 ↑0.3 |
| TLVS-Refine | ✓ | × | 88.71 ↑3.62 | 4.1 ↓7.9 | 44.5 ↓5.8 | 28.1 ↓8.3 | 80.8 ↑6.1 |
| TLVS | ✓ | ✓ | 89.78 ↑4.69 | 3.8 ↓8.2 | 51.8 ↑1.5 | 19.7 ↓16.7 | 82.2 ↑7.5 |
| **Qwen2.5-VL-7B** | | | | | | | |
| Vanilla | – | – | 86.90 | 5.4 | 51.6 | 29.0 | 89.4 |
| TLVS-Init | × | × | 86.83 ↓0.07 | 5.0 ↓0.4 | 50.8 ↓0.8 | 26.8 ↓2.2 | 87.6 ↓1.8 |
| TLVS-Refine | ✓ | × | 87.00 ↑0.10 | 4.2 ↓1.2 | 48.9 ↓2.7 | 20.0 ↓9.0 | 88.8 ↓0.6 |
| TLVS | ✓ | ✓ | 87.33 ↑0.43 | 3.8 ↓1.6 | 51.2 ↓0.4 | 16.7 ↓12.3 | 89.8 ↑0.4 |

88.71 and reduces AMBER hallucinations, at a small cost in coverage. Adding token-level adaptive steering, the full TLVS further reduces hallucinations while recovering coverage to 51.8 and improving F1 from 80.8 to 82.2. A consistent pattern is observed on Qwen2.5-VL-7B: refinement reduces hallucinations but lowers coverage, whereas the full TLVS recovers coverage to 51.2 and further decreases hallucinations.

### 5.4. Mechanistic and Qualitative Analyses

TLVS produces a stepwise steering strength $\lambda_t \in [\lambda_{\min}, \lambda_{\max}]$ from the visual-sensitivity signal (Eq. (17)–(22)). To verify that TLVS steers where it matters, we analyze how the adaptive schedule distributes steering budget across token types and sample groups. We divide tokens into visually critical and visually uncritical ones, and further split the visually critical tokens into hallucinated vs. non-hallucinated ones based on annotations of the final generated output. For simplicity, we leverage COCO annotations and CHAIR (Rohrbach et al., 2018) to map these three groups to `hall_obj`, `correct_obj`, and `other` tokens, respectively. More specifically, we stratify samples by the number of object mentions in the generated output and analyze each group separately.

As shown in Fig. 8, first, visually critical tokens receive substantially higher adaptive strength than the overall average (`all`), indicating that TLVS concentrates budget on steps that require visual grounding; second, within visually critical tokens, hallucinated tokens receive higher adaptive strength than non-hallucinated tokens, suggesting that the adaptive schedule concentrates interventions on hallucination-prone steps rather than uniformly perturbing the full sequence, consistent with our token-level sparsity diagnosis. Finally, `has_hall` samples tend to receive higher adaptive strength than `no_hall`, though residual hallucinations can remain due to bounded strength and imperfect step selection. Together, these findings provide evidence that TLVS reallocates steering budget toward visually crit-

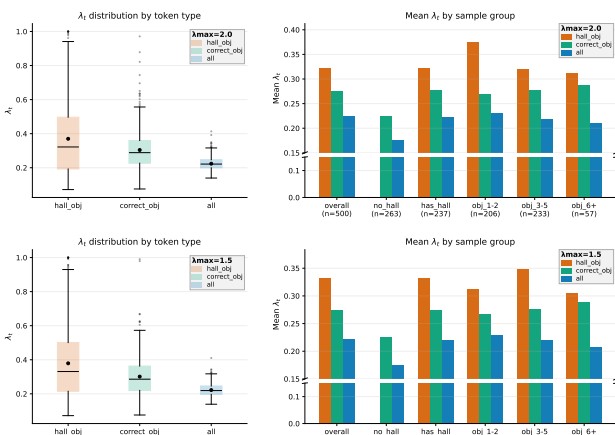

*Figure 8.* Adaptive steering focuses on critical tokens. $\lambda_t$ distribution (left) and mean $\lambda_t$ by token group (right).

*Table 3.* Efficiency and Overhead. (i) Fwd/Token, the average number of forward passes per generated token; (ii) ms/Token, wall-clock latency per token; (iii) Sec/Sample, end-to-end generation time per sample; and (iv) Peak VRAM, the maximum GPU memory footprint during inference. Lower is better for all four metrics (↓).

| Method | Fwd/Token ↓ | ms/Token ↓ | Sec/Sample ↓ | Peak VRAM (GB) ↓ |
|---|---|---|---|---|
| Base | 1.00 | 31.25 | 3.34 | 13.95 |
| TLVS | 2.00 | 64.01 | 6.79 | 14.03 |
| Overhead (Ours/Base) | 2.00× | 2.05× | 2.04× | 1.01× |

ical steps and reduces collateral perturbations on image-insensitive tokens.

### 5.5. Efficiency and Overhead

We measure inference-time overhead on the COCO captioning evaluation set, using the same prompts and decoding settings as in our main experiments. As shown in Table 3, TLVS introduces a predictable ~2× inference-time overhead, since it uses an auxiliary contrastive route similar to two-branch decoding methods (e.g., VCD (Leng et al., 2024)). Notably, the peak VRAM increase is negligible (1.01×), making TLVS a practical, training-free reliability knob for improving grounding and factuality without additional parameters or repeated sampling.

## 6. Conclusion

We study activation steering in LVLMs and find that image conditioning is sparse and heavy-tailed across decoding steps, making step-invariant global steering low-SNR for direction extraction and inefficient in budget allocation. Motivated by this, we propose Token-Level Visual-Sensitivity Steering (TLVS), which extracts token-level steering vectors, optionally refines them with lightweight corrections, and applies step-adaptive steering only where it matters. Across POPE, AMBER, CHAIR (COCO), MMHal-Bench,

and HallusionBench, TLVS consistently reduces hallucinations on both LLaVA-1.5-7B and Qwen2.5-VL-7B while largely preserving coverage and generation quality.

## Acknowledgements

This work was funded in part by the National Natural Science Foundation of China under Grant No. 62536004, in part by the STI2030-Major Projects grant from the Ministry of Science and Technology of the People's Republic of China under Grant No. 2021ZD0200700, in part by the Key-Area Research and Development Program of Guangdong Province under Grant No. 2023B0303030001, and in part by the Science and Technology Program of Guangzhou under Grant No. 2024A04J6310.

## Impact Statement

This work aims to reduce hallucinations in large vision-language models by improving the visual faithfulness of generated responses. It may help make multimodal systems more reliable in applications such as visual question answering, image captioning, and assistive technologies. However, our method does not eliminate hallucinations completely and should not be regarded as a guarantee of factual or perceptual correctness. In high-stakes scenarios, model outputs should still be carefully verified by humans or external validation systems.

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

# A. Details of Observation 1

We detail the token-level visual-sensitivity analysis used in Figure 3 and Figure 4. Our goal is to quantify, at each decoding step, how much the presence of an image changes the model's likelihood assigned to the same reference response under teacher forcing. The overall procedure is summarized in Algorithm 1. Unless otherwise specified, we run this analysis on the AMBER benchmark (Wang et al., 2023).

**AMBER protocol and annotations.**  AMBER provides images with structured annotations and an LLM-free evaluation protocol covering existence, attribute, and relation hallucinations (Wang et al., 2023). Following AMBER's generative setting, we obtain the reference response $y_{1:T}$ (Wang et al., 2023).

**Reference response and token range.**  Given an input sample $x$ (including the prompt and image, when applicable), we first obtain a fixed reference response $y_{1:T}$ by running the model with vanilla decoding (i.e., without steering). All token-level statistics are computed on the *answer tokens* $y_{1:T}$ only, excluding prompt tokens and any special tokens outside the response span.

**Teacher-forced per-token log-probabilities.**  For each step $t$, we compute the next-token log-probability under teacher forcing as

$$\log p(y_t \mid x, \cdot, y_{<t}),  \tag{23}$$

where $\cdot$ indicates the conditioning variant (image-conditioned vs. image-ablated). In practice, we obtain all steps from a single forward pass over the concatenated sequence (prompt + response) and read the log-probability assigned to the reference token $y_t$ at each position.

**Image-conditioned vs. image-ablated inputs.**  We construct two matched conditions that share the identical text and response tokens: (i) an image-conditioned input $(x, I)$ and (ii) an image-ablated (or weakly conditioned) input $(x, \varnothing)$. Crucially, we keep the textual prefix $y_{<t}$ identical in both conditions for every step $t$, so that the measured difference reflects image conditioning rather than sampling-path divergence. For models that require explicit image placeholder tokens, we keep the same placeholder structure (e.g., the same number of image tokens) while removing the visual content in the ablated variant.

**Token-level visual sensitivity.**  We quantify token-level visual sensitivity via the teacher-forced log-likelihood difference:

$$\Delta_t = \log p(y_t \mid x, I, y_{<t}) - \log p(y_t \mid x, \varnothing, y_{<t}),  \tag{24}$$

which matches Eq. (3) in the main text. By construction, $\Delta_t$ is token-aligned across the two conditions. We visualize per-example $\Delta_t$ along decoding steps using lollipop plots, and highlight large-magnitude steps by thresholding $|\Delta_t| \geq \tau$.

**Dataset-level aggregation and relative position.**  To aggregate across varying response lengths, we associate each answer token with its relative position $r_t = t/T \in (0, 1]$. The hexbin density plot in Figure 4 is produced by pooling pairs $(r_t, \Delta_t)$ over all examples and plotting the log-scaled bin counts (i.e., $\log_{10}(\text{count})$) to reveal the concentration near $\Delta_t \approx 0$ and the sparse heavy-tailed deviations.

**Concentration metrics (top-$p$ mass).**  To characterize how much of the total $|\Delta_t|$-mass is carried by a small fraction of steps, for each response we rank tokens by $|\Delta_t|$ and compute the cumulative mass captured by the top fraction $p$:

$$S(p) = \frac{\sum_{t \in \text{Top}(p)} |\Delta_t|}{\sum_t |\Delta_t|},  \tag{25}$$

where $\text{Top}(p)$ denotes the set of the top $p$ fraction of tokens ranked by $|\Delta_t|$. We compute $S(p)$ per response and report the dataset average. We report $S(p)$ at representative values $p \in \{1, 5, 10, 20, 50, 100\}\%$ and visualize the resulting top-heavy accumulation pattern in Figure 3.

**Implication for steering direction extraction.**  The observed sparsity and mass concentration indicate that a small set of visually sensitive steps dominates the image-vs-no-image difference signal. Therefore, methods that average activation differences uniformly over all steps (as in Eq. (1)) can dilute the contribution of these critical steps, yielding a lower signal-to-noise steering direction.

---

**Algorithm 1** Token-level visual sensitivity via teacher forcing

---

**Require:** sample $x$, image $I$, ablated image $\varnothing$, LVLM $p_\theta$
 1: Decode a reference response $y_{1:T}$ with vanilla decoding (no steering).
 2: Compute $\log p(y_t \mid x, I, y_{<t})$ for all $t$ under teacher forcing.
 3: Compute $\log p(y_t \mid x, \varnothing, y_{<t})$ for all $t$ under teacher forcing.
 4: $\Delta_t \leftarrow \log p(y_t \mid x, I, y_{<t}) - \log p(y_t \mid x, \varnothing, y_{<t})$.
 5: Record $(r_t = t/T, \Delta_t)$ for hexbin; compute $S(p)$ by ranking $|\Delta_t|$ within the response.

---

## B. Theory for Observation 1: signal dilution under sparse visual influence

We formalize why uniformly averaging activation differences across all steps can dilute the visual signal when image influence is sparse, and why this issue is exacerbated when the two conditions generate different sequences.

**Setup.** Fix a layer $\ell$ and define the step-wise activation difference $\delta_{\ell,t}(x) = h_{\ell,t}^I(x) - h_{\ell,t}^\varnothing(x) \in \mathbb{R}^d$. Let $\mathcal{C} \subseteq \{1, \ldots, T\}$ denote the (unknown) set of visually critical steps and let $q = |\mathcal{C}|/T$.

**Assumption B.1** (Sparse-effect model). There exists a vector $v_\star^{(\ell)} \in \mathbb{R}^d$ such that

$$\mathbb{E}[\delta_{\ell,t}(x) \mid t \in \mathcal{C}] = v_\star^{(\ell)}, \qquad \mathbb{E}[\delta_{\ell,t}(x) \mid t \notin \mathcal{C}] = 0,$$

and the per-step noise is zero-mean with bounded second moment: $\mathbb{E}\big[\|\delta_{\ell,t}(x) - \mathbb{E}\delta_{\ell,t}(x)\|_2^2\big] \leq \sigma^2$.

**Proposition B.2** (Dilution of token-averaged differences). *Under Assumption B.1, the token-averaged estimator used by step-invariant baselines,*

$$\hat{v}_{avg}^{(\ell)} = \frac{1}{T} \sum_{t=1}^{T} \delta_{\ell,t}(x),$$

*has expectation*

$$\mathbb{E}\big[\hat{v}_{avg}^{(\ell)}\big] = q\, v_\star^{(\ell)}.$$

*Moreover, consider the token-selected estimator*

$$\hat{v}_{\mathcal{C}}^{(\ell)} = \frac{1}{|\mathcal{C}|} \sum_{t \in \mathcal{C}} \delta_{\ell,t}(x),$$

*which satisfies $\mathbb{E}[\hat{v}_{\mathcal{C}}^{(\ell)}] = v_\star^{(\ell)}$. If one rescales the averaged estimator to be unbiased, $\tilde{v}_{avg}^{(\ell)} = \hat{v}_{avg}^{(\ell)}/q$, then its variance is worse by a factor on the order of $1/q$:*

$$\mathbb{E}\big[\|\tilde{v}_{avg}^{(\ell)} - v_\star^{(\ell)}\|_2^2\big] \lesssim \frac{1}{q} \mathbb{E}\big[\|\hat{v}_{\mathcal{C}}^{(\ell)} - v_\star^{(\ell)}\|_2^2\big].$$

*Proof.* By linearity of expectation and Assumption B.1,

$$\mathbb{E}\big[\hat{v}_{avg}^{(\ell)}\big] = \frac{1}{T} \sum_{t=1}^{T} \mathbb{E}[\delta_{\ell,t}(x)] = \frac{|\mathcal{C}|}{T} v_\star^{(\ell)} = q\, v_\star^{(\ell)}.$$

Similarly, $\mathbb{E}[\hat{v}_{\mathcal{C}}^{(\ell)}] = v_\star^{(\ell)}$. For the variance comparison, under the bounded second-moment condition, averaging over $n$ steps yields mean-square error scaling as $\mathcal{O}(\sigma^2/n)$ (up to constant factors depending on temporal dependence). Thus,

$$\mathbb{E}\big[\|\hat{v}_{\mathcal{C}}^{(\ell)} - v_\star^{(\ell)}\|_2^2\big] \lesssim \frac{\sigma^2}{|\mathcal{C}|} = \frac{\sigma^2}{qT},$$

while

$$\mathbb{E}\big[\|\tilde{v}_{avg}^{(\ell)} - v_\star^{(\ell)}\|_2^2\big] = \frac{1}{q^2} \mathbb{E}\big[\|\hat{v}_{avg}^{(\ell)} - \mathbb{E}\hat{v}_{avg}^{(\ell)}\|_2^2\big] \lesssim \frac{1}{q^2} \cdot \frac{\sigma^2}{T} = \frac{\sigma^2}{q^2 T}.$$

Taking the ratio gives an $\mathcal{O}(1/q)$ gap, completing the proof. □

**Remark: additional path-divergence noise without token alignment.** The above argument assumes that $\delta_{\ell,t}(x)$ isolates the effect of image conditioning at comparable steps. In practice, when the image-conditioned and image-ablated conditions *free-run* and generate different sequences, $\delta_{\ell,t}(x)$ additionally absorbs a *path-divergence* term induced by different textual contexts, which can further reduce the effective SNR of token-averaged differences. This motivates using token-aligned diagnostics (e.g., teacher forcing) to estimate visually critical steps, and performing token-aware selection/weighting when extracting steering directions.

# C. Details of Observation 2

We detail the teacher-forced, token-aligned diagnosis used in Figure 5. The goal is to quantify, at each decoding step, how different intervention schedules redistribute likelihood over an identical reference response. Given an input sample $x$ (including the prompt and image, when applicable), we first obtain a fixed reference response $y_{1:T}$ by running the model with vanilla decoding (no steering). We then evaluate per-token NLLs on the same $y_{1:T}$ under teacher forcing for each schedule, ensuring token-level alignment across conditions.

**Per-token NLL under teacher forcing.** For each step $t$, we compute

$$\mathrm{NLL}_t = -\log p(y_t \mid x, y_{<t}), \tag{26}$$

by feeding the identical prefix $y_{<t}$ and reading the log-probability assigned to the next reference token $y_t$.[1] This yields a token-aligned likelihood profile over the same response $y_{1:T}$ for any intervention schedule.

**Intervention schedules.** We compare two schedules that share the same steering direction and intervened layer set: **(i) Global step-invariant steering**, which applies a constant steering strength $\lambda_{\mathrm{const}}$ at every decoding step; and **(ii) Oracle token-aware steering** (diagnosis only), which uses AMBER annotations to define a set of visually critical steps $\mathcal{C}$ and applies a larger strength $\lambda_{\mathrm{high}}$ for $t \in \mathcal{C}$ and a smaller strength $\lambda_{\mathrm{low}}$ otherwise. The oracle schedule is only used for diagnosis to probe the limitation of step-invariant steering and is not used at test time.

**Token alignment for visually critical steps.** We map AMBER-labeled spans (hallucination and object-related spans) to token indices of the reference response $y_{1:T}$ to obtain $\mathcal{C}$. Concretely, we tokenize $y_{1:T}$ and use the tokenizer's offset mapping to align annotated character spans to token steps. A token step $t$ is included in $\mathcal{C}$ if its token span overlaps with any annotated hallucination or object-related span.

**Computing $\Delta\mathrm{NLL}_t$.** For each schedule, we obtain $\mathrm{NLL}_t^{\mathrm{method}}$ under teacher forcing and compute

$$\Delta\mathrm{NLL}_t = \mathrm{NLL}_t^{\mathrm{method}} - \mathrm{NLL}_t^{\mathrm{vanilla}}, \tag{27}$$

as in Eq. (6). Positive $\Delta\mathrm{NLL}_t$ indicates suppression of the reference token $y_t$ (lower probability), while negative values indicate promotion.

# D. Implementation Details

### D.1. Implementation and hardware.

All experiments are implemented in PyTorch with the Hugging Face transformers ecosystem. We run training and evaluation on a single NVIDIA A100 GPU. We load the base LVLMs checkpoints from their corresponding Hugging Face repositories and follow the official tokenizer and image preprocessing pipelines provided with each model. Our code and configuration files will be released to facilitate reproducibility.

### D.2. Hyperparameter settings

We report TLVS hyperparameters for LLaVA-1.5-7B and Qwen2.5-VL-7B; unless noted, the protocol is shared across models.

---

[1]In practice, we obtain all $\mathrm{NLL}_t$ from a single forward pass over the concatenated (prompt + response) sequence and read the next-token log-probability at each position.

**Step 1 (pseudo-labeling).** We set $q_{\text{tail}} = 0.95$ on $|\Delta_t|$ (Eq. 8), taking top/bottom $5\%$ as $\mathcal{T}^+/\mathcal{T}^-$. Induced thresholds: Qwen2.5-VL-7B ($|\Delta_t| \geq 3.0$, $|\Delta_t| \leq 0.125$); LLaVA-1.5-7B ($|\Delta_t| \geq 2.3$, $|\Delta_t| \leq 0.435$).

**Step 2 (vector refinement).** We optimize only $W^{(\ell)}$ on layers $\mathcal{S}$ (model-specific) with Eq. 13 using $\beta = 0.05$ and $\gamma = 0.10$. AdamW: lr $5 \times 10^{-4}$, wd 0, clip 1.0, grad-accum 8, 1500 micro-steps.

**Step 3 (adaptive steering).** We compute $\text{VS}_t$ with $\tau_{\text{KL}} = 1.0$ (Eq. 17), normalize with $(\mu_{\text{VS}}, \sigma_{\text{VS}}) = (0.0913, 0.293)$ (Eq. 18), and gate with $(b, s) = (2.25, 1.0)$ (Eq. 19). We set $(\lambda_{\min}, \lambda_{\max}) = (0, 1.5)$ (Eq. 20) and disable smoothing by default ($\alpha = 0$ in Eq. 21).

**Per-step confidence cap.** To avoid overly aggressive interventions at steps where the model is already confident, we optionally impose a per-step cap $\lambda_t^{\text{cap}}$ based on a lightweight confidence proxy derived from the image-conditioned route. By default we use a margin-based proxy and set the cap mode to `margin`, with cap upper bound $\lambda_{\text{cap}} = 2.0$. This cap is applied after the gating computation and we take $\lambda_t = \min(\hat{\lambda}_t, \lambda_t^{\text{cap}})$ as in Eq. 22. All cap-related calibration statistics and implementation constants are reported in our released configuration for exact reproducibility.

**Layer choices.** We steer and refine all transformer layers except the first and the last. For Qwen2.5-VL-7B we use layers 1–26. For LLaVA-1.5-7B we use layers 1–30.

## E. Per-step confidence cap for adaptive steering

We further constrain the stepwise steering strength by imposing a per-step cap $\lambda_t^{\text{cap}}$ computed from a confidence proxy of the image-conditioned logits $z_t^I$. The cap is designed to prevent overly aggressive interventions at steps where the model is already confident, thereby improving decoding stability.

**Image-conditioned predictive distribution and entropy.** At decoding step $t$, let $p_t^I$ denote the next-token distribution under the image-conditioned route:

$$p_t^I = \text{softmax}\left(\frac{z_t^I}{T_{KL}}\right), \tag{28}$$

where $T_{KL}$ is the temperature used consistently with the visual-sensitivity computation in Eq. (17). We measure confidence using the (Shannon) entropy of $p_t^I$:

$$H_t = -\sum_{v \in \mathcal{V}} p_t^I(v) \log p_t^I(v), \tag{29}$$

where $\mathcal{V}$ is the vocabulary. Lower entropy indicates higher confidence.

**Entropy normalization.** To make entropy comparable across steps and samples, we standardize it using calibration statistics computed on a held-out calibration set:

$$\bar{H}_t = \frac{H_t - \mu_H}{\sigma_H + \epsilon}, \tag{30}$$

We use the same $\epsilon$ as in Eq. (18), a small constant for numerical stability (e.g., $\epsilon = 10^{-8}$), where $(\mu_H, \sigma_H)$ denote the mean and standard deviation of entropy values pooled over calibration tokens.

**Mapping entropy to a steering cap.** We map the normalized entropy $\bar{H}_t$ to a cap in $[\lambda_{\min}, \lambda_{\max}]$ via a sigmoid:

$$c_t = \text{sigmoid}\left(\frac{\bar{H}_t - b_H}{s_H}\right), \tag{31}$$

and define the per-step cap as

$$\lambda_t^{\text{cap}} = \lambda_{\min} + (\lambda_{\max} - \lambda_{\min})\, c_t. \tag{32}$$

Intuitively, when the model is more uncertain (higher entropy), $c_t$ increases and the cap permits stronger steering; when the model is confident (lower entropy), the cap becomes smaller and prevents unnecessary perturbations.

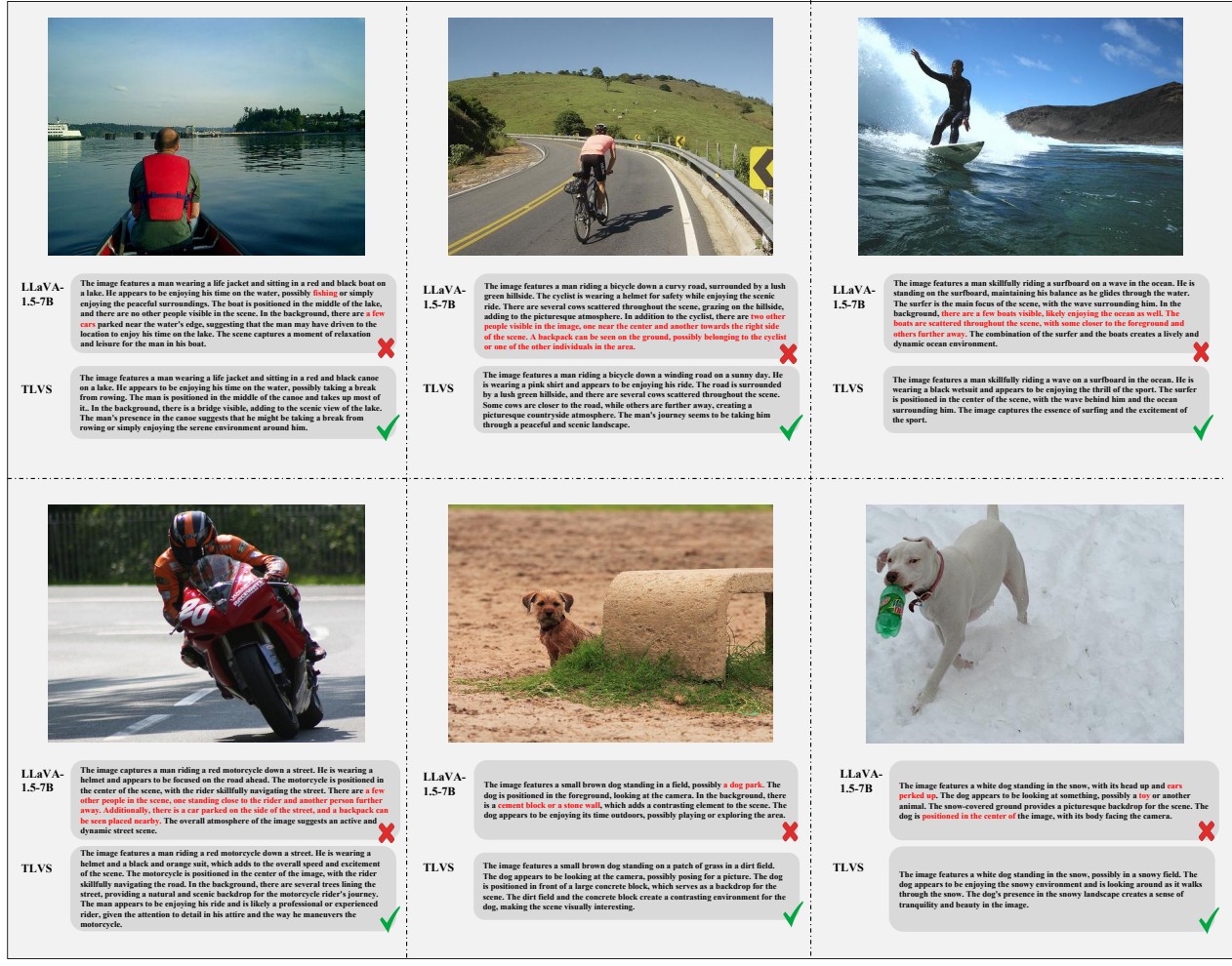

*Figure 9.* **Qualitative comparison before vs. after TLVS.** We compare vanilla decoding (without steering) against TLVS on four representative cases. TLVS reduces hallucinated object mentions and improves visual grounding, while avoiding broad perturbations on image-insensitive tokens due to step-adaptive steering.

**Final capped strength.** During decoding, we apply the cap after optional smoothing (Eq. (21)) and take

$$\lambda_t = \min(\hat{\lambda}_t, \lambda_t^{\text{cap}}), \tag{33}$$

as in Eq. (22).

## F. Qualitative comparison examples

We provide qualitative examples comparing base model and the model with our Token-Level Visual-Sensitivity Steering (TLVS). Each example is given a general prompt such as "Describe this image". Overall, TLVS suppresses ungrounded object mentions while preserving visually supported content, consistent with its step-adaptive budget allocation.

## G. Data Requirement for Learning Token-Level Visual-Sensitivity Patterns

The steering vectors in TLVS are not intended to model the case-level distribution of RLHF-V. Instead, RLHF-V image-question pairs are used to capture token-level activation patterns that distinguish visually sensitive tokens from visually

*Table 4.* Pseudo-label classifier scaling study on RLHF-V subsets. Even small subsets are sufficient to capture the token-level visual-sensitivity pattern.

| $N$ | Micro-F1 | Micro-Bal-Acc | AUROC |
|-----|----------|---------------|-------|
| 50  | 0.8244   | 0.8792        | 0.9585 |
| 100 | 0.8475   | 0.9004        | 0.9672 |
| 150 | 0.8518   | 0.9076        | 0.9699 |
| 200 | 0.8605   | 0.9141        | 0.9734 |
| 250 | 0.8675   | 0.9186        | 0.9762 |
| 300 | 0.8721   | 0.9226        | 0.9791 |
| 350 | 0.8811   | 0.9288        | 0.9813 |
| 400 | 0.8887   | 0.9331        | 0.9827 |
| 450 | 0.8946   | 0.9372        | 0.9849 |
| 500 | 0.9008   | 0.9410        | 0.9863 |

insensitive tokens. Specifically, for each answer token, we compute the visual-sensitivity score

$$\Delta_t = \log p_\theta(y_t \mid x, I, y_{<t}) - \log p_\theta(y_t \mid x, \varnothing, y_{<t}), \tag{34}$$

where the two probabilities are computed under teacher forcing with and without the image input. Based on $|\Delta_t|$, we construct pseudo-labeled token groups $\mathcal{T}^+$ and $\mathcal{T}^-$, corresponding to visually sensitive and visually insensitive tokens, respectively.

To examine how much data is required to learn this token-level pattern, we conduct a pseudo-label classifier scaling study on RLHF-V subsets with different numbers of samples. The classifier is trained to distinguish visually sensitive tokens from visually insensitive tokens, and is evaluated on a fixed 500-sample subset. As shown in Table 4, even a small number of samples is sufficient to learn the token-level "visual vs. non-visual" pattern. Increasing the subset size further improves the classification performance, mainly by improving stability rather than changing the learned pattern.

These results indicate that the token-level visual-sensitivity pattern can be learned from a relatively small calibration subset. This supports our design choice that the refinement stage is primarily used to denoise and stabilize the steering vectors, rather than to memorize the case-level distribution of RLHF-V.

