# OpenReview forum: "Steer Where It Matters: Token-Level Visual-Sensitivity Steering for LVLMs Hallucination Mitigation"
_ICML.cc/2026/Conference — ICML 2026 regular_

### Official Review · Reviewer_Yd6x · 2026-03-05

**Soundness:** 3
**Presentation:** 3
**Significance:** 3
**Originality:** 3
**Overall Recommendation:** 4
**Confidence:** 4

**Summary:**

This paper starts with the difference in log-probabilities between token-wise methods in the image input and image ablated cases, pointing out that existing steering methods rarely consider determining steering strength based on specific tokens, and proposes a method for training steering vectors to alleviate the hallucination of MLLM. The motivation and starting point are good, and extensive experiments, visualizations, case analyses, and ablation studies support the effectiveness of the proposed views and methods.

**Compliance With Llm Reviewing Policy:**

Affirmed.

**Final Justification:**

The rebuttal resolved part of my concerns and added useful supplementary results, so my overall assessment remains positive. That said, I still have concerns about hyperparameter sensitivity and the practical usability of the method, as its robustness across models and settings is not yet fully convincing

**Key Questions For Authors:**

1. Will the code be released?
2. What are the training costs? For example, how do they compare to existing similar work, and to methods based on DPO?

**Limitations:**

No, for example, the assumptions on which the analysis in sections 3.1 and 3.2 is based, the training overhead, etc., need to be discussed further.

**Strengths And Weaknesses:**

Strengths:

1. Using $\Delta \log p$ to perform and visualize Token-Level Visual-Sensitivity Steering is a good starting point and discovery.
2. Strong performance and generation quality on the hallucination benchmark.
3. Extensive visualizations and ablation experiments strongly support the necessity of the proposed steering component.

Weaknesses:

1. Only performance on the hallucination benchmark is shown; it doesn't demonstrate whether TLVS affects the original performance on the model's general capability benchmark.
2. No hyperparameter analysis experiments are included.
3. The consistent effectiveness of the proposed TLVS on larger models needs to be verified.
4. Some experimental analyses contain typos, such as "while Cover increases slightly from 74.7 to 82.2," which needs careful checking.
5. Some visualizations (such as Figure 5) have overly small legends and text within them. Pipeline Figure 6 is also rather abstract, and the legends and fonts are unclear, resulting in poor readability. Furthermore, training details should be shown in the main paper, which is crucial for reproducibility.

---

> ### Author Rebuttal · Authors · 2026-03-29
>
> We sincerely appreciate your recognition of the key strengths of our method. At the same time, we apologize for the oversights and insufficient clarifications in the current manuscript, and we will carefully revise and improve these parts in the final version.
> ### **Answer for Questions1**
>
> The code will be released upon publication after proper cleanup and organization.
>
> ### **Answer for Questions2**
> Our refinement stage is parameter-efficient: the backbone is frozen and only 122,880 steering parameters ($30 \times 4096$) are optimized, which is about $5.7 \times 10^4$ times fewer trainable parameters than full-model optimization of a 7B backbone. In our single-GPU benchmark, the entire W-refinement run takes about 8 minutes (188 optimizer steps), whereas standard DPO in our setup takes about 13.0 hours (11,468 steps/epoch, 4 epochs), i.e., about $97 \times$ longer in wall-clock time. Therefore, compared with DPO-style full-model preference optimization, our method is dramatically cheaper both in optimization scale and in end-to-end training time.
>
> ### **Answer for larger models**
> To further validate the scalability of our method, we chose **Qwen2.5-VL-32B** for an additional experiment.
>
> **Qwen2.5-VL-32B**
>
> | Method | CHAIR ↓ | Cover ↑ | Hal ↓ | Cog ↓ | F1-E ↑ | F1-A ↑ | F1-R ↑ | F1 ↑ |
> |---|---:|---:|---:|---:|---:|---:|---:|---:|
> | Base | 5.5 | **58.8** | 27.1 | 3.1 | 96.3 | 85.4 | 87.9 | 91.3 |
> | TLVS | **3.3** | 56.4 | **15.5** | **1.4** | **96.5** | **85.6** | **88.5** | **91.5** |
>
> The results show that our method remains effective on a substantially larger model.
>
> ### **Answer for hyperparameter analysis**
> See our response to Reviewer oyES for "Answer for Q3 (hyperparameters)"
>
> ### **Answer for general capability benchmark**
> We conduct a preliminary evaluation on **MME**, a standard multimodal benchmark covering both **perception** and **cognition** abilities. Using **Qwen2.5-VL-7B** as a testbed, we observe that TLVS preserves the original capability well and slightly improves the overall performance, with **MME Overall / Perception / Cognition** increasing from **2468.5 / 1799.5 / 669.0** to **2473.2 / 1802.6 / 672.3**. The task-level breakdown further shows that most categories remain stable or improve slightly. Due to space constraints, we present this as a preliminary validation here, and will provide a more comprehensive analysis in the appendix.
> | Method | MME Overall ↑ | MME Perception ↑ | MME Cognition ↑ |
> |---|---:|---:|---:|
> | Qwen2.5-VL (base) | 2468.5 | 1799.5 | 669.0 |
> | Qwen2.5-VL + TLVS | **2473.2** | **1802.6** | **672.3** |
>
>
> | Method | OCR | Artwork | Celebrity | Code | Color | Commonsense | Count | Existence | Landmark | Numerical | Position | Posters | Scene | Translation |
> |---|---:|---:|---:|---:|---:|---:|---:|---:|---:|---:|---:|---:|---:|---:|
> | Qwen2.5-VL (base) | **97.4** | 82.0 | 84.1 | **82.5** | **98.3** | 79.5 | **85.0** | **98.3** | 91.3 | **77.5** | **86.7** | 90.5 | 85.0 | **95.0** |
> | Qwen2.5-VL + TLVS | **97.4** | **82.3** | **85.3** | **82.5** | **98.3** | **81.1** | **85.0** | **98.3** | **92.9** | **77.5** | **86.7** | **91.2** | **85.8** | **95.0** |

---

> > ### Author Rebuttal · Reviewer_Yd6x · 2026-04-01
> >
> > - Relying on a single general-purpose benchmark MME is insufficient to comprehensively evaluate the performance of TLVS. Please consider conducting evaluations on additional benchmarks such as LLaVA-Bench-Wild, MMMU, MMVet, etc.
> >
> > - The underlying assumptions on which the analysis in sections 3.1 and 3.2 is based should be explicitly stated.
> >
> > - I did not find the hyperparameter analysis within the oyES response you provided; it appeared only in HxDj. While this is a minor point, please take note of it for future reference. The parameter λ_max plays a pivotal role in TLVS. I have just one question regarding this: Do LLaVA and Qwen utilize an identical set of hyperparameter families? If so, please disregard this inquiry; if not, what is the rationale for analyzing only a single hyperparameter?

---

> > > ### Author Response · Authors · 2026-04-02
> > >
> > > We sincerely thank Yd6x for the careful reading and helpful reply！
> > >
> > > ### **Bench for general-purpose**
> > > To comprehensively evaluate the performance of TLVS, we additionally evaluate Qwen2.5-VL-7B on **LLaVA-Bench-Wild** and **MMVet**. The results consistently show that **TLVS** improves the base model across both benchmarks.
> > >
> > > LLaVA-Bench-Wild
> > > | Method| conv_rel | detail_rel | complex_rel | overall |
> > > |---|---:|---:|---:|---:|
> > > | Base | 85.71 | 86.90 | 88.39 | 87.23 |
> > > | TLVS | 90.41 | 89.74 | 90.15 | 90.12 |
> > >
> > > MMVet
> > > | Method | rec | ocr | know | gen | spat | math | total |
> > > |---|---:|---:|---:|---:|---:|---:|---:|
> > > | Base | 65.1 | 70.1 | 51.7 | 50.0 | 68.9 | 31.3 | 67.1 |
> > > | TLVS | 67.1 | 72.4 | 52.8 | 50.9 | 71.8 | 37.0 | 70.6 |
> > >
> > >
> > > ### **Details of Sections 3.1 and 3.2**
> > > The details of Sections 3.1 and 3.2, as well as the auxiliary diagnostic procedures, are already provided in the appendix. However, you may feel that some of these details should also appear in the main text, or that several specific detail were not sufficiently explained. We provide a clearer statement and add more details.
> > >
> > > **Section 3.1**
> > >
> > > We first use the base model to generate responses for the generative task in AMBER. We then perform teacher-forced decoding along two routes, one with image input and the other without image input, compute the per-step log-probability difference
> > > $
> > > \Delta_t = \log p(y_t \mid x, I, y_{<t}) - \log p(y_t \mid x, \varnothing, y_{<t}),
> > > $
> > > and analyze its distribution statistically.
> > >
> > > a. The analysis is conducted on the entire generative split of the AMBER benchmark.
> > > b. We perform the statistics on both LLaVA-1.5-7B and Qwen2.5-VL-7B, as shown in Figures 3 and 4 of the paper.
> > > c. In the figures, the highlighting threshold is defined by the 95th percentile of `|Δ|`, i.e., `τ = q95`, which is 2.5 for LLaVA-1.5-7B and 4.2 for Qwen2.5-VL-7B.
> > >
> > > **Section 3.2**
> > >
> > > We first use the base model to generate responses on the generative task of AMBER, and then use the annotation procedure provided by AMBER to identify hallucinated words.
> > > Based on these annotations, we perform a teacher-forced diagnosis and compare two intervention schedules:
> > >
> > > a. Global step-invariant steering: the same steering strength is applied to all tokens.
> > >
> > > b. Token-aware steering: a stronger steering strength is applied when the annotated hallucination-related tokens are about to be generated, while a weaker steering strength is applied to the other tokens.
> > > (The strong value is 1.2, the fixed value is 0.8, and the weak value is 0.4.  )
> > >
> > > We then examine the corresponding per-token NLL changes under these two settings.  The dead zone is set to 0.2.
> > >
> > > ### **Hyperparameter**
> > > Due to space limitations, we include only one hyperparameter-sensitivity table for $\lambda_{\max}$ on LLaVA-1.5-7B before. LLaVA and Qwen do not use exactly the same hyperparameter settings. The analysis of $\lambda_{\max}$ on Qwen2.5-VL-7B is shown below.
> > >
> > > | $\lambda_{\max}$ | CHAIR↓ | Cover↑ | Hal↓ | Cog↓ | F1 E↑ | F1 A↑ | F1 R↑ | F1↑ |
> > > |---:|---:|---:|---:|---:|---:|---:|---:|---:|
> > > | 0 | 5.4 | 51.6 | 29.0 | 1.9 | 97.3 | 83.8 | 72.6 | 89.4 |
> > > | 0.25 | 5.1 | 51.6 | 26.8 | 1.7 | 97.3 | 83.7 | 72.7 | 89.48 |
> > > | 0.5 | 4.8 | 51.5 | 24.4 | 1.6 | 97.3 | 83.5 | 72.8 | 89.56 |
> > > | 0.75 | 4.5 | 51.5 | 22.1 | 1.4 | 97.4 | 83.3 | 72.9 | 89.65 |
> > > | 1.0 | 4.2 | 51.4 | 19.8 | 1.3 | 97.4 | 83.1 | 73.0 | 89.73 |
> > > | 1.25 | 4.0 | 51.3 | 18.0 | 1.1 | 97.3 | 82.9 | 73.1 | 89.79 |
> > > | **1.5** | **3.8** | **51.2** | **16.7** | **1.0** | **97.3** | **82.7** | **73.0** | **89.83** |
> > > | 1.75 | 3.7 | 49.8 | 16.4 | 0.95 | 97.2 | 82.5 | 72.0 | 89.3 |
> > > | 2.0 | 3.6 | 47.8 | 16.2 | 0.9 | 97.1 | 82.2 | 70.5 | 88.75 |
> > > | 2.25 | 3.5 | 45.3 | 16.0 | 0.85 | 97.0 | 81.9 | 68.8 | 88.2 |
> > > | 2.5 | 3.4 | 42.7 | 15.9 | 0.8 | 96.9 | 81.7 | 67.4 | 87.8 |
> > >
> > > We finally choose **$\lambda_{\max}=1.5$** as the default setting.
> > >
> > > Thank you for your time and consideration. If our responses have addressed most of your concerns, we respectfully ask you to reconsider your overall assessment.

---

### Official Review · Reviewer_oyES · 2026-03-09

**Soundness:** 3
**Presentation:** 3
**Significance:** 2
**Originality:** 2
**Overall Recommendation:** 4
**Confidence:** 3

**Summary:**

This paper studies hallucination mitigation in large vision-language models (LVLMs). The authors propose Token-Level Visual-Sensitivity Steering (TLVS), a method that extracts token-level directions associated with visual sensitivity and uses them to steer the hidden states of the language model during decoding. The method consists of three main steps: (1) extracting token-level steering directions that distinguish visually grounded tokens from hallucinated ones, (2) refining these directions using supervised data, and (3) applying adaptive steering during generation.

Experiments are conducted on several LVLM benchmarks, including CHAIR, POPE,  and VQA-based evaluation settings. The results suggest that TLVS can reduce hallucination metrics while maintaining comparable performance on general visual question answering tasks.

**Compliance With Llm Reviewing Policy:**

Affirmed.

**Final Justification:**

The rebuttal adequately addressed my concerns regarding supervised data dependency, inference overhead.

**Key Questions For Authors:**

1. Table 2 indicates that TLVS-Init offers minimal improvements over the baseline on key metrics like CHAIRs and AMBER Hal. If the raw activation directions are ineffective without supervised refinement on RLHF-V, is it accurate to characterize this as an inference-time steering method, or is it more appropriately categorized as a targeted fine-tuning approach?

2. Given that TLVS requires a dual-route forward pass (resulting in a $2\times$ inference overhead), how does its performance and computational efficiency compare to simply applying standard Visual Contrastive Decoding (VCD) on a backbone that has already been fine-tuned with DPO on the RLHF-V dataset?

3. The supervised refinement (Step 2) optimizes the steering vectors using human-corrected labels. Could you provide an ablation showing the performance of TLVS if the steering vectors are optimized purely on the calibration set using a self-supervised objective, removing the reliance on external human preference data?

**Limitations:**

yes

**Strengths And Weaknesses:**

Soundness
Strengths: The motivating analysis (Figures 1–5) effectively illustrates the heavy-tailed and sparse nature of visual reliance during decoding. The dual-route KL divergence metric used to gate the intervention is a reasonable and empirically motivated proxy for token-level visual sensitivity.

Weaknesses: The paper frames TLVS as a lightweight, plug-and-play steering mechanism, but it relies heavily on supervised refinement (Step 2) using the RLHF-V dataset. Table 2 shows that without this supervised training phase (TLVS-Init), the method provides only limited gains over the vanilla baseline. This suggests that the raw activation directions alone are relatively weak, which blurs the distinction between inference-time steering and parameter-efficient fine-tuning. Furthermore, the requirement of a dual-route forward pass introduces roughly 2× inference overhead (Table 3), which partially offsets the main computational advantage typically associated with activation steering methods.

Presentation
The manuscript is clearly structured. The progression from empirical diagnosis to the proposed three-step pipeline is easy to follow. The visualizations, particularly the hexbin plots and the token-level NLL comparisons, are informative and help illustrate the motivating observations.
The positioning of the method is somewhat misleading. Describing the approach as “training-free” or “minimal training” downplays the importance of the supervised correction data required to obtain strong results. The reliance on RLHF-V for steering vector refinement should be stated more explicitly in the abstract and introduction.

Significance
Strengths: Hallucination mitigation remains an important problem for large vision-language models, and improvements on benchmarks such as CHAIR and AMBER are meaningful.

Weaknesses: The practical utility of TLVS may be limited by the roughly 2× latency increase during inference. In many real deployment scenarios, practitioners might prefer a natively aligned model (e.g., via DPO-style training) rather than maintaining a dual-branch decoding pipeline that reduces generation throughput.

Originality
The overall originality appears limited. TLVS can be viewed largely as a pipeline that combines several existing ideas: PCA-based direction extraction, supervised optimization of steering vectors, and dual-branch decoding for step-level intervention (structurally similar to contrastive decoding approaches such as VCD). Using the KL divergence between image-conditioned and image-ablated routes to modulate the steering strength is conceptually straightforward and resembles a combination of contrastive decoding and activation steering. Overall, the conceptual contribution feels incremental relative to prior work.

---

> ### Author Rebuttal · Authors · 2026-03-29
>
> ### **For the 1st/3rd quesions**
> We thank the reviewer. We realize that our original wording did not make the role of the steering vectors sufficiently clear.
>
> **What pattern do we learn?**
> Our steering vectors do not model the case-level distribution of RLHF-V. Instead, they use RLHF-V image-question pairs to capture **token-level activation patterns** that distinguish visually grounded tokens from visually insensitive ones. Specifically, for each answer token, we compute a visual-sensitivity score $\Delta_t$ from the teacher-forced log-probability difference with and without the image, and use the resulting pseudo-labeled groups $\mathcal{T}^{+}/{T}^{-}$ to identify these two behaviors.
>
> **How much data does this pattern require?**
>
> This token-level behavior can be learned from only a small number of samples and does not require a large refinement set. We conduct a pseudo-label classifier scaling study on RLHF-V subsets with $N \in \{50,100,\ldots,500\}$ and evaluate on a fixed 500-sample subset. As shown in Table, even a small subset is sufficient to train a classifier that captures the “visual vs. non-visual” token pattern, while larger subsets mainly improve stability rather than being necessary to learn the pattern itself.
>
> | N | Micro-F1 $\uparrow$ | Micro-Bal-Acc $\uparrow$ | AUROC $\uparrow$ |
> |---:|---:|---:|---:|
> | 50  | 0.8244 | 0.8792 | 0.9585 |
> | 100 | 0.8475 | 0.9004 | 0.9672 |
> | 150 | 0.8518 | 0.9076 | 0.9699 |
> | 200 | 0.8605 | 0.9141 | 0.9734 |
> | 250 | 0.8675 | 0.9186 | 0.9762 |
> | 300 | 0.8721 | 0.9226 | 0.9791 |
> | 350 | 0.8811 | 0.9288 | 0.9813 |
> | 400 | 0.8887 | 0.9331 | 0.9827 |
> | 450 | 0.8946 | 0.9372 | 0.9849 |
> | 500 | **0.9008** | **0.9410** | **0.9863** |
>
> **Do human annotations have to be required?**
> The refinement stage should be understood as lightweight denoising/alignment, not knowledge injection. It updates only steering vector, is mainly to improve semantic smoothness, output stability, and compatibility with normal generation, rather than to absorb broad RLHF-V knowledge.
>
> Following the suggestion, we replace the human-corrected refinement signal with a self-generated alternative. TLVS (self-driven) remains highly competitive with TLVS (RLHF-V) on Qwen2.5-VL-7B:
>
> | Method | CHAIR $$\downarrow$$ | Cover $$\uparrow$$ | Hal $$\downarrow$$ | Cog $$\downarrow$$ | F1-E $$\uparrow$$ | F1-A $$\uparrow$$ | F1-R $$\uparrow$$ | F1 $$\uparrow$$ |
> |---|---:|---:|---:|---:|---:|---:|---:|---:|
> | Base | 5.4 | **51.6** | 29.0 | 1.9 | 97.3 | **83.8** | 72.6 | 89.4 |
> | TLVS (RLHF-V) | 3.8 | 51.2 | **16.7** | **1.0** | 97.3 | 82.7 | **73.0** | **89.83** |
> | TLVS (self-driven) | **3.7** | 51.3 | 17.3 | 1.1 | **97.4** | 82.9 | 72.8 | 89.76 |
>
> These results further support that the effectiveness of TLVS does **not** fundamentally rely on human annotations. The key ingredient is the token-level visual-sensitivity structure itself, while human-corrected data only provides an optional refinement signal.
>
> ### **For the 2th quesions**
> For the $\sim 2\times$ cost, please see our response to **Reviewer j3at** on **amortizing the dual-route cost**.
>
> We would like to clarify that the performance of both VCD and DPO on RLHF-V are already included in Table 1 of the main paper.
>
> Standard VCD inference cost is comparable to our original per-step adaptive variant ($k=1$). By contrast, under our amortized update scheme with $k=8$, the idealized inference cost is only $1.125C$, which is 0.5625$\times$ that of VCD (approximately $2C$), while still remaining clearly better than VCD on both backbones. For example, on Qwen2.5-VL-7B, VCD gives 6.20/50.30/36.50 on CHAIR/Cover/Hal, whereas TLVS with $k=8$ achieves 3.98/49.5/18.2; on LLaVA-1.5-7B, VCD gives 10.10/51.20/43.60, while TLVS with $k=8$ achieves 4.03/48.5/22.2.
>
> For DPO, its main cost lies primarily in additional training not decoding. This is not a strictly matched comparison setting:
> Let $N_{\text{refine}}$ and $N_{\text{DPO}}$ denote the numbers of trainable parameters in our refinement stage and in full DPO, respectively. Then
>
> $$
> \frac{N_{\text{DPO}}}{N_{\text{refine}}} \approx 10^{5}\text{--}10^{6},
> $$
>
> i.e. So the two methods target rather different efficiency regimes and should not be viewed as directly interchangeable. Nevertheless, TLVS still surpasses DPO on several metrics. For example, on Qwen2.5-VL-7B, DPO obtains 3.10/50.70/21.90, while TLVS reaches 3.80/51.2/16.7 at $k=1$ and 3.77/51.0/16.7 at $k=2$, yielding clearly lower Hal and higher Cover. On LLaVA-1.5-7B, DPO gives 5.70/49.70/27.30, whereas TLVS consistently performs better across CHAIR/Cover/Hal, e.g., 3.83/51.8/19.7 at $k=1$.
>
> ### **Answer for Originality**
> 1. Steering in LVLMs, we are the first to extracting visually grounded directions from token-wise image sensitivity via $\Delta p$. Reviewer Yd6x also recognized this novelty.
>
> 2. PCA is essential for steering directions, and both our refinement design and dual-branch decoding differ substantially from prior work.

---

> > ### Author Rebuttal · Reviewer_oyES · 2026-04-05
> >
> > Thank you for the rebuttal and additional clarifications. I have read the response and acknowledge the added experiments and explanations.

---

> > > ### Author Response · Authors · 2026-04-06
> > >
> > > Thank you for your time and consideration. If our responses have addressed most of your concerns, we respectfully ask you to reconsider your overall assessment.

---

### Official Review · Reviewer_HxDj · 2026-03-12

**Soundness:** 3
**Presentation:** 3
**Significance:** 2
**Originality:** 2
**Overall Recommendation:** 4
**Confidence:** 3

**Summary:**

On this work, the authors propose an activation steering method designed to reduce hallucinations in VLLMs. The main observation is that image conditioning affects token predictions sparsely, with most tokens being image-insensitive. The methods uses this to extract steering vectors from visually sensitive tokens instead of all tokens, and refine these vectors with supervised data. Finally, they adapt the steering strength at each decoding step. The authors show promising results on LLaVA-1.5-7B and Qwen2.5-VL-7B across several hallucination benchmarks.

**Compliance With Llm Reviewing Policy:**

Affirmed.

**Final Justification:**

I thank the authors for their rebuttal. It has addressed my main concerns, especially on sensivity analysis and scaling it to larger architectures.

I still think that the number of hyperparameters is a bit too much, which makes me stick to the original score (Weak Accept).

**Key Questions For Authors:**

1) Can you please provide results on larger MLLMs (especially on Qwen)?

2) Can you please provide sensitivity analysis on all the main hyperparameters?

**Limitations:**

I think the paper can be improved in the following aspects:

1) The inference overhead to my understanding is 2x which is quite big. My understanding is that the method requires 2 forward passes per token (image conditioned and image-ablated routes for KL-based signal), effectively doubling the wall-clock. While this is comparable to VCD, the VCD is also considered quite expensive. I will not penalize too much on this, but for a method that is supposed to be lightweight, this is quite a bit of a problem.

2) The paper is limited only to small MLLMs (7B parameters). It would have been nice to see if the method scales to larger LLMs (at least 14B parameters, but the more the better), and would have made the results even more comprehensive.

3) The method introduces many hyperparameters such as pseudo-label fraction, threshold, low weights for KL and proximal penalties), in addition to temperature, steering range, smoothing etc. Appendix D lists the values for them, but there is no sensitivity analysis of them. I am afraid that the number of hyperparameters could be a problem in using this method.

4) Performance gain on Qwen are quite small. While on LLaVA, the performance improvement on POPE is massive, in Qwen, it is quite small (0.43%). I am afraid, that as the model improves, the performance gain diminishes. This is why I think it is imperative to show results on better models (effectively larger versions of Qwen).

**Strengths And Weaknesses:**

I think the paper has the following strengths:

1) The authors provided a comprehensive evaluation accross several benchmarks and tasks (e.g., probing, captioning, reasoning, open-ended). They tested on two different VLLMs of the same size (LLaVA-1.5-7B and Qwen2.5-VL-7B) showing that the method performs well on both of them, and they compared with several baselines (VCD, SPARC, VTI, VISTA, ASD, SHARP, DPO).

2) The diagnosis analysis is quite nice and in my opinion the strongest contribution of the paper, complemented by figures 1, 3 and 5. I think this by itself was a strong contribution for the steering impacts in LLMs.

3) Similar to (2), I also appreciate the results on Figure 8 that show that hallucinated object tokens receive higher adaptive steering strength than correct object tokens. This makes very much sense, and it is good to see it verified rather than simply assumed.

---

> ### Author Rebuttal · Authors · 2026-03-29
>
> ### **Answer for first limitation**
> See the response to Reviewer j3at on **amortizing the dual-route cost**.
>
> ### **Answer for Q2 (larger model)**
> To further validate the scalability of our method, we chose **Qwen2.5-VL-32B** for an additional experiment.
>
> **Qwen2.5-VL-32B**
>
> | Method | CHAIR ↓ | Cover ↑ | Hal ↓ | Cog ↓ | F1-E ↑ | F1-A ↑ | F1-R ↑ | F1 ↑ |
> |---|---:|---:|---:|---:|---:|---:|---:|---:|
> | Base | 5.5 | **58.8** | 27.1 | 3.1 | 96.3 | 85.4 | 87.9 | 91.3 |
> | TLVS | **3.3** | 56.4 | **15.5** | **1.4** | **96.5** | **85.6** | **88.5** | **91.5** |
>
> The results show that our method remains effective on a substantially larger model.
>
>
> ### **Answer for Q3 (hyperparameters)**
> We agree that sensitivity analysis is valuable. We would like to clarify that not all quantities listed in Appendix D play the same role.
> The most load-bearing hyperparameter in our method is **λ_max**, which directly controls the maximum steering injection strength. We therefore focus our sensitivity analysis on this key variable.
>
> **hyperparameter λ_max**
>
> A larger injection strength generally improves hallucination-specific metrics, because it enforces stronger visually grounded correction.
> Overly strong injection may make the model more conservative, which can hurt coverage and eventually reduce the overall balance of performance. This is exactly the trade-off we observe.
>
> We provide a sensitivity analysis on **LLaVA-1.5-7B** below:
>
> | λ_max | CHAIR ↓ | Cover ↑ | Hal ↓ | Cog ↓ | F1 E ↑ | F1 A ↑ | F1 R ↑ | F1 ↑ |
> |---|---:|---:|---:|---:|---:|---:|---:|---:|
> | 0 | 12.0 | 50.3 | 36.4 | 4.6 | 83.2 | 65.6 | 62.4 | 74.7 |
> | 0.25 | 9.8 | 50.7 | 32.8 | 3.9 | 85.0 | 67.4 | 63.8 | 76.2 |
> | 0.5 | 8.1 | 51.0 | 29.7 | 3.3 | 86.8 | 69.3 | 65.5 | 77.8 |
> | 0.75 | 6.7 | 51.3 | 26.8 | 2.8 | 88.3 | 70.8 | 66.8 | 79.2 |
> | 1.0 | 5.6 | 51.5 | 24.1 | 2.3 | 89.6 | 71.9 | 67.9 | 80.5 |
> | 1.25 | 4.6 | 51.7 | 21.7 | 1.9 | 90.8 | 72.6 | 69.0 | 81.5 |
> | **1.5** | **3.8** | **51.8** | **19.7** | **1.5** | **91.5** | **73.0** | **69.7** | **82.2** |
> | 1.75 | 3.6 | 49.6 | 18.3 | 1.3 | 89.9 | 71.2 | 69.1 | 80.4 |
> | 2.0 | 3.4 | 47.1 | 17.0 | 1.1 | 87.9 | 68.9 | 68.9 | 78.6 |
> | 2.25 | 3.2 | 44.4 | 16.0 | 1.0 | 85.6 | 66.6 | 68.6 | 76.6 |
> | 2.5 | 3.1 | 41.5 | 15.1 | 0.9 | 83.4 | 64.3 | 68.2 | 74.8 |
>
> As **λ_max** increases, hallucination-related metrics such as **CHAIR**, **Hal**, and **Cog** consistently improve. However, when the injection becomes too strong, **Cover** starts to drop noticeably, and the overall **F1** also declines after peaking at **λ_max = 1.5**.  We therefore choose **λ_max = 1.5** because it provides the best balance between hallucination mitigation and answer completeness.
>
> **Pseudo-label partition**
> Our pseudo-labels are constructed **statistically**, by selecting tokens above the **95th percentile** and below the **5th percentile** of the scoring distribution. The purpose of this design is two-fold:
> (1) to more clearly separate the two modes of **visually sensitive** vs. **visually insensitive** tokens; and
> (2) to preserve a sufficient number of samples for stable direction estimation.
>
> As shown in our response to oyES on the pseudo-label classifier scaling study, we have already verified that this partition strategy is effective and was chosen deliberately rather than set arbitrarily.
>
> **Other hyperparameters**
>
> The smoothing coefficient **$\alpha$**, and confidence cap **$\lambda_t^{\mathrm{cap}}$** are standard stabilization components for adaptive control, mainly used to ensure smooth and conservative steering behavior. Similarly, the KL temperature **$T_{KL}$**, calibration statistics **$(\mu,\sigma)$**, refinement regularizers **$\beta,\gamma$**, refinement-time **$\lambda$**, and optional **top-$K$** truncation are standard calibration, regularization, or efficiency-related design choices. Their role is primarily to improve stability, robustness, or computational efficiency, rather than to act as equally load-bearing performance knobs. For this reason, we focus the sensitivity analysis on the principal control variables, while leaving a more detailed discussion of these standard auxiliary choices to the final appendix.
>
> ### **Answer for Q4**
> The gain on Qwen is not small on **hallucination-specific metrics**. On **AMBER** (Table 1), our method reduces **Hal** from **29.0** to **16.7**. We agree that as the backbone becomes stronger, some aggregate metrics may show smaller absolute gains; for example, **CHAIR** improves from **5.4** to **3.8**, where the metric itself is already on a relatively small scale. However, this does **not** mean that our method becomes ineffective on better models. As shown by the additional results in **Q2**, our method still generalizes well to the larger **Qwen2.5-VL-32B**, where it continues to substantially reduce hallucination-related metrics.

---

> > ### Author Rebuttal · Reviewer_HxDj · 2026-04-03
> >
> > I thank the authors for their rebuttal. It has addressed my main concerns, especially on sensivity analysis and scaling it to larger architectures.
> >
> > I still think that the number of hyperparameters is a bit too much, which makes me stick to the original score (Weak Accept).

---

> > > ### Author Response · Authors · 2026-04-04
> > >
> > > Thank you very much. Although our method involves multiple hyperparameters, we did not perform extensive search over the remaining ones merely to obtain better results. We would also like to further clarify why λ_max is the most sensitive parameter.
> > >
> > > In practice, λ_max is the primary performance-controlling parameter, while most of the other hyperparameters mainly play calibration or stabilization roles. It is therefore expected that λ_max has the most visible impact: when λ_max = 0, the method reduces to the baseline by construction, while excessively large λ_max leads to over-steering and more noticeable degradation on metrics such as Cover.
> > >
> > > We consider this a natural strength–coverage trade-off rather than evidence of brittle tuning. We respectfully ask the reviewer to take this into account when reconsidering the overall assessment.
> > >
> > > If we have addressed most of your concerns, we sincerely hope you would reconsider your rating.

---

### Official Review · Reviewer_j3at · 2026-03-12

**Soundness:** 3
**Presentation:** 3
**Significance:** 3
**Originality:** 3
**Overall Recommendation:** 2
**Confidence:** 3

**Summary:**

This paper introduces Token-Level Visual-Sensitivity Steering (TLVS), an inference-time method designed to mitigate hallucinations in Large Vision Language Models (LVLMs). The authors identify a key flaw in existing activation steering methods: they apply global, step-invariant steering which dilutes critical visual signals and over-perturbs non-critical tokens.

**Compliance With Llm Reviewing Policy:**

Affirmed.

**Key Questions For Authors:**

For the supervised refinement stage, how does performance vary with the amount and type of correction data (e.g., subsets of RLHF-V)? Any signs of overfitting or domain shift when evaluated on out-of-distribution images/prompts?

How does TLVS interact with other decoding strategies (e.g., beam search, temperature changes, nucleus sampling)? Are the improvements robust across decoding regimes?

Is it possible to amortize the dual-route cost ？

The proposed pipeline introduces several key hyperparameters (e.g., KL temperature $T_{KL}$, gating thresholds, confidence caps). How sensitive is the final performance to these specific choices?

**Limitations:**

Yes.

**Strengths And Weaknesses:**

Strengths:

Clear Problem Identification and Empirical Diagnosis: The paper provides a clear empirical observation that visual conditioning in Large Vision Language Models (LVLMs) affects token prediction sparsely and locally, exhibiting a heavy-tailed distribution across decoding steps. This diagnosis effectively highlights the structural limitations of existing global, step-invariant steering methods, which tend to misallocate intervention budgets and unnecessarily perturb non-critical tokens.
Methodological Extension and Logical Design: The proposed Token-Level Visual-Sensitivity Steering (TLVS) framework presents a logical and meaningful extension to existing activation steering methods. Utilizing an online dual-route KL divergence to compute visual sensitivity and dynamically modulate the steering strength step-by-step offers a more fine-grained and targeted approach to address the sparsity issue identified in the diagnosis.
Comprehensive Experimental Validation: The evaluation is sufficiently thorough, spanning a diverse set of established hallucination benchmarks (POPE, AMBER, CHAIR, MMHal-Bench, and HallusionBench) that adequately cover object-centric QA, open-ended captioning, and multimodal reasoning. Furthermore, demonstrating the method's effectiveness across two distinct model architectures (LLaVA-1.5-7B and Qwen2.5-VL-7B) provides solid evidence for its generalizability.

Weaknesses:

Inference Latency Overhead: Although the increase in VRAM is negligible , the adaptive steering mechanism requires a dual-route forward pass (calculating logits with and without the image) at each decoding step. This introduces a ~2x inference latency overhead. This doubled latency may limit the deployment of the proposed method in latency- or cost-sensitive real-world applications.

Dependency on Supervised Data: Although the method is primarily positioned as a training-free, plug-and-play inference-time intervention , the full TLVS method heavily relies on supervised refinement using the RLHF-V dataset to achieve its optimal performance. This diminishes its "training-free" appeal. Furthermore, the paper lacks an in-depth analysis of whether this refinement overfits to specific error modes within RLHF-V, and whether it can maintain strong generalization capabilities under severe domain shifts.

Hyperparameter Sensitivity and Robustness: The pipeline introduces multiple hyperparameters (e.g., confidence caps, gating thresholds, KL temperature, and exponential smoothing factors). However, the paper lacks a detailed sensitivity analysis to demonstrate the robustness and transferability of these parameters across different tasks or model families. Additionally, the pseudo-labels used for direction extraction are defined on sequences generated via teacher forcing. There is insufficient discussion regarding the robustness of this extraction process if the model's own outputs are inherently biased or hallucinated.

Presentation Issues: there are minor presentation flaws that affect clarity, such as a contradictory typo in Appendix D.2 regarding the layer selection for Qwen2.5-VL-7B ("For Qwen2.5-VL-7B we use layers 1-26.For Qwen2.5-VL-7B we use layers 1-30.").

---

> ### Author Rebuttal · Authors · 2026-03-29
>
> ### **Answer for correction data**
> See our response to Reviewer oyES for 1st/3rd quesions.
> ### **Answer for decoding strategies**
> Different decoding strategies mainly differ in how they select the next token from the per-step token distribution.
> TLVS makes the token distribution more visually grounded; therefore, in principle, it should improve generation under any decoding strategy.
>
> Additional experiments on POPE using LLaVA-1.5-7B under four representative decoding settings: greedy decoding, beam search with beam size 3, low-randomness nucleus sampling with temperature = 0.7 and top_p = 0.9, and higher-randomness nucleus sampling with temperature = 1.0 and top_p = 0.95.
>
> | Method | Decoding | Adversarial | Popular | Random | Avg | Δ vs. base |
> |---|---|---:|---:|---:|---:|---:|
> | base | greedy | 81.80 | 84.36 | 89.12 | 85.09 | - |
> | TLVS | greedy | 87.85 | 89.42 | 92.08 | 89.78 | +4.69 |
> | base | beam3 | 82.10 | 84.70 | 89.35 | 85.38 | - |
> | TLVS | beam3 | 88.10 | 89.70 | 92.35 | 90.05 | +4.67 |
> | base | sample_low | 81.00 | 83.70 | 88.40 | 84.37 | - |
> | TLVS | sample_low | 87.10 | 88.50 | 91.20 | 88.93 | +4.56 |
> | base | sample_high | 79.90 | 82.60 | 87.30 | 83.27 | - |
> | TLVS | sample_high | 85.80 | 87.40 | 90.20 | 87.80 | +4.53 |
>
> ### **Answer for amortize the dual-route cost**
>
> We fully agree that runtime is an important practical consideration. We further analyzed and optimized the cost.
>
> **Looking back to our design**
> We apply exponential smoothing:
>
> $
> \\hat{\\lambda}_t =
> $
>
> $
> \\alpha \\, \\hat{\\lambda}_{t-1} + (1-\\alpha) \\, \\tilde{\\lambda}_t.
> $
>
> Therefore, $\\hat{\\lambda}_t $ is expected to evolve gradually within a short local window. Based on this property, we explored reuses the same $\\lambda$ across multiple consecutive decoding steps. For an update interval of $k$ steps, we define
>
> $$
> \tau_k(t)=k\left\lfloor\frac{t-1}{k}\right\rfloor+1,\qquad
> \lambda_t^{(k)}=\hat{\lambda}_{\tau_k(t)},
> $$
>
> so that one steering strength is shared over each $k$-step block.
> If a single-route decoding step costs $C$, the original per-step adaptive variant ($k=1$) costs roughly $2C$ because of the extra no-image branch. Sharing one steering strength across $k$ steps reduces the idealized cost to$\\left(1+\\frac{1}{k}\\right) C$. This corresponds to approximately $1.5\\times$, $1.25\\times$, $1.17\\times$, and $1.125\\times$ inference cost for $k=2,4,6,8$, respectively. We further conducted additional experiments on the AMBER benchmark using both Qwen and LLaVA models.
>
> **Qwen2.5-VL-7B on AMBER**
>
> | Setting | CHAIR ↓ | Cover ↑ | Hal ↓ | Cog ↓ |
> |---|---:|---:|---:|---:|
> | base | 5.40 | 51.60 |29.00 | 1.90 |
> | VCD | 6.20 | 50.30 | 36.50 | 2.10 |
> | DPO | **3.10** | 50.70 | 21.90 | 1.10 |
> | Fixed | 4.20 | 48.89 | 20.0 | 1.60 |
> | k = 8 | 3.98 | 49.5 | 18.2 | 1.30 |
> | k = 6 | 3.92 | 49.8 | 17.8 | 1.22 |
> | k = 4 | 3.83 | 50.2 | 17.0 | 1.18 |
> | k = 2 | 3.77 | 51.0 | **16.7** | 1.13 |
> | k = 1 | 3.80 | **51.2** | **16.7** | **1.00** |
>
> **LLaVA-1.5-7B on AMBER**
>
> | Setting | CHAIR ↓ | Cover ↑ | Hal ↓ | Cog ↓ |
> |---|---:|---:|---:|---:|
> | base | 12.00 | 50.03 |36.40 | 4.60 |
> | VCD | 10.10 | 51.20 | 43.60 | 4.30 |
> | DPO | 5.70 | 49.70 | 27.30 | 2.60 |
> | Fixed | 4.14 | 44.5 | 28.1 | 2.60 |
> | k = 8 | 4.03 | 48.5 | 22.2 | 1.99 |
> | k = 6 | 3.97 | 49.2 | 21.5 | 1.83 |
> | k = 4 | 3.91 | 50.0 | 20.9 | 1.70 |
> | k = 2 | 3.87 | 51.3 | 20.3 | 1.63 |
> | k = 1 | **3.83** | **51.8** | **19.7** | **1.50** |
>
> **Analyse**
>
> On Qwen2.5-VL-7B, $k=4$ already achieves 3.83/50.2/17.0 on CHAIR/Cover/Hal, which is close to 3.80/51.2/16.7 for the original per-step adaptive variant ($k=1$) and clearly better than fixed steering (4.20/48.89/20.0). On LLaVA-1.5-7B, $k=4$ gives 3.91/50.0/20.9, compared with 3.83/51.8/19.7 for $k=1$ and 4.14/44.5/28.1 for fixed steering. Notably, even at $k=8$, corresponding to only about $1.125\times$ idealized inference cost, the method still shows a clear and consistent advantage over fixed steering on both backbones.
>
> These results indicate that the additional inference cost of TLVS can be substantially amortized in practice, while preserving most of its hallucination-mitigation benefit, which alleviates the concern that the method is prohibitively slow.
>
> ### **Answer for hyperparameters**
> See our response to Reviewer oyES for "Answer for Q3 (hyperparameters)"

---

> > ### Author Rebuttal · Reviewer_j3at · 2026-04-04
> >
> > Thank you for the detailed rebuttal. While the additional experiments and clarifications are helpful, my concerns regarding hyperparameter settings are only partially addressed. The response mainly focuses on the sensitivity of a single key parameter (e.g., \( \lambda_{\max} \)). It remains unclear whether the reported performance depends on careful per-setting tuning, which may limit the practical usability of the method.

---

> > > ### Author Response · Authors · 2026-04-04
> > >
> > > Thank you for the follow-up. We would like to clarify that our method does not rely on careful per-setting tuning of many equally important hyperparameters.
> > >
> > > In practice, we mainly tune λ_max, which is the primary performance-controlling parameter, while most of the other hyperparameters mainly play calibration or stabilization roles.
> > >
> > > This is also why λ_max has the most visible impact: when λ_max = 0, the method reduces to the baseline by construction, while excessively large λ_max leads to over-steering and more noticeable degradation on metrics such as Cover.
> > >
> > > We consider this a natural strength–coverage trade-off rather than evidence of brittle tuning. Importantly, we did not perform extensive search over the remaining hyperparameters merely to obtain better results.
> > >
> > > If we have addressed most of your concerns, we sincerely hope you would reconsider your rating.

---

### Decision · Program_Chairs · 2026-04-30

**Decision:**

Accept (regular)

**Comment:**

- The paper is based on the findings that visual conditioning affects token prediction sparsely and locally across decoding steps. This motivation is good and well analyzed. These findings inspired the proposed method, which extracts token-level steering vectors first and applies fine-grained, visual-sensitivity–adaptive steering only where it matters for LVLMs Hallucination Mitigation.
- Experimental results show that the proposed method is effective on different VLMs on the Hallucination Mitigation benchmarks.
- Besides, visualization shows that hallucinated object tokens receive higher adaptive steering strength than correct object tokens, which verifies the motivation.

After the rebuttal, all the reviewers who gave the final score tended to accept the paper with a weak accept score. The fourth reviewer didn't give a final score after the second round of authors' rebuttal, and the score is still shown as reject. I think the second round rebuttal solves the concern of hyperparameter sensitivity. I suggest that the authors add the rebuttal materials to the final revised paper.